# Human-interpretable image features derived from densely mapped cancer pathology slides predict diverse molecular phenotypes

James A. Diao [1,2,5], Jason K. Wang[1,2,5], Wan Fung Chui [1,2,5], Victoria Mountain[1], Sai Chowdary Gullapally[1], Ramprakash Srinivasan[1], Richard N. Mitchell [2,3], Benjamin Glass[1], Sara Hoffman[1], Sudha K. Rao[1], Chirag Maheshwari[1], Abhik Lahiri[1], Aaditya Prakash[1], Ryan McLoughlin[1], Jennifer K. Kerner[1], Murray B. Resnick[1,4], Michael C. Montalto[1], Aditya Khosla[1], Ilan N. Wapinski[1], Andrew H. Beck [1,5 ✉], Hunter L. Elliott[1,5] & Amaro Taylor-Weiner[1,5 ✉]

Computational methods have made substantial progress in improving the accuracy and throughput of pathology workflows for diagnostic, prognostic, and genomic prediction. Still, lack of interpretability remains a significant barrier to clinical integration. We present an approach for predicting clinically-relevant molecular phenotypes from whole-slide histopathology images using human-interpretable image features (HIFs). Our method leverages >1.6 million annotations from board-certified pathologists across >5700 samples to train deep learning models for cell and tissue classification that can exhaustively map whole-slide images at two and four micron-resolution. Cell- and tissue-type model outputs are combined into 607 HIFs that quantify specific and biologically-relevant characteristics across five cancer types. We demonstrate that these HIFs correlate with well-known markers of the tumor microenvironment and can predict diverse molecular signatures (AUROC 0.601–0.864), including expression of four immune checkpoint proteins and homologous recombination deficiency, with performance comparable to 'black-box' methods. Our HIF-based approach provides a comprehensive, quantitative, and interpretable window into the composition and spatial architecture of the tumor microenvironment.

[1] PathAI, Inc., Boston, MA, USA. [2] Program in Health Sciences and Technology, Harvard Medical School, Boston, MA, USA. [3] Department of Pathology, Brigham & Women's Hospital, Harvard Medical School, Boston, MA, USA. [4] Department of Pathology, Warren Alpert Medical School, Providence, RI, USA. [5] These authors contributed equally: James A. Diao, Jason K. Wang, Wan Fung Chui, Andrew H. Beck, Hunter L. Elliott, Amaro Taylor-Weiner. ✉email: andy.beck@pathai.com; amaro.taylor@pathai.com

While manual microscopic inspection of histopathology slides remains the gold standard for evaluating the malignancy, subtype, and treatment options for cancer[1], pathologists and oncologists increasingly rely on molecular assays to guide personalization of cancer therapy[2]. These assays can be expensive and time-consuming[3] and, unlike histopathology images, are not routinely collected, limiting their use in retrospective and exploratory research. Manual histological evaluation, on the other hand, presents several clinical challenges. Careful inspection requires significant time investment by board-certified anatomic pathologists and is often insufficient for prognostic prediction. Several evaluative tasks, including diagnostic classification, have also reported low inter-rater agreement across experts and low intra-rater agreement across multiple reads by the same expert[4,5]. Furthermore, manual assessment of the expression of specific genes from histopathology has not to our knowledge been demonstrated.

Modern computer vision methods present the potential for rapid, reproducible, and cost-effective clinical and molecular predictions. Over the past decade, the quantity and resolution of digitized histology slides has dramatically improved[6]. At the same time, the field of computer vision has made significant strides in pathology image analysis[7,8], including automated prediction of tumor grade[9], mutational subtypes[10], and gene expression signatures across cancer types[11–13]. In addition to achieving diagnostic sensitivity and specificity metrics that match or exceed those of human pathologists[14–16], automated computational pathology can also scale to service resource-constrained settings where few pathologists are available. As a result, there may be opportunities to integrate these technologies into the clinical workflows of developing countries[17].

However, end-to-end deep learning models that infer outputs directly from raw images present significant risks for clinical settings, including fragility of machine learning models to population shift between training and real-world application, technical variability in sample preparation and analysis, and other unpredictable failure modes[18–20]. Many of these risks stem from lack of interpretability of "black-box" models[21,22]. "Black-box" model predictions are difficult for users to interrogate and understand, leading to user distrust and inability to diagnose failure modes or identify reliance on confounding correlates. Without reliable means for understanding when and how vulnerabilities may become failures, computational methods may face difficulty achieving widespread adoption in clinical settings[23,24].

One emerging solution has been the automated computation of human-interpretable image features (HIFs) to predict clinical outcomes. HIF-based prediction models often mirror the pathology workflow of searching for distinctive, stage-defining features under a microscope and offer opportunities for pathologists to validate intermediate steps and identify failure points. In addition, HIF-based solutions enable incorporation of histological knowledge and expert pixel-level annotations, which increases predictive power. Studied HIFs span a wide range of visual features, including stromal morphological structures[25], cell and nucleus morphologies[26], shapes and sizes of tumor regions[27], tissue textures[28], and the spatial distributions of tumor-infiltrating lymphocytes (TILs)[29,30].

In recent years, the relationship between the tumor microenvironment (TME) and patient response to targeted therapies has been made increasingly clear[31,32]. For instance, immuno-supportive phenotypes, which exhibit greater baseline antitumor immunity and improved immunotherapy response, have been linked to the presence of TILs and elevated expression of programmed death-ligand 1 (PD-L1) on tumor-associated immune cells. In contrast, immuno-suppressive phenotypes have been linked to the presence of tumor-associated macrophages and fibroblasts, as well as reduced PD-L1 expression[32–34]. HIF-based approaches have the potential to provide an interpretable window into the composition and spatial architecture of the TME in a manner complementary to conventional genomic approaches. While prior HIF-based studies have identified many useful feature classes, most have been limited in scope. Studies to date often involve a single cell or tissue type; none have explored features that combine both cell and tissue properties. In addition, the majority of reported HIFs have only been vetted on a single cancer type, often non-small cell lung cancer (NSCLC).

In this research study, we present a computational pathology pipeline that can integrate high-resolution cell- and tissue-level information from whole-slide images (WSIs) to predict treatment-relevant, molecularly derived phenotypes across five different cancer types. Our approach combines the predictive power of deep learning with the interpretability of HIFs, which enables explicit incorporation of prior knowledge and achieves performance comparable to end-to-end models. We introduce a diverse collection of 607 HIFs ranging from simple cell (e.g., density of lymphocytes in cancer tissue) and tissue quantities (e.g., area of necrotic tissue) to complex spatial features capturing tissue architecture, tissue morphology, and cell–cell proximity. In this study, we demonstrate that such features can generalize across cancer types and provide a quantitative and interpretable link to specific and biologically relevant characteristics of each TME.

## Results

**Dataset characteristics and fully automated pipeline design.** In order to test our approach on a diverse array of histopathology images, we obtained 2917 hematoxylin and eosin (H&E)-stained, formalin-fixed, and paraffin-embedded (FFPE) WSIs from The Cancer Genome Atlas (TCGA), corresponding to 2634 distinct patients. These images, each scanned at either ×20 or ×40 magnification, represented patients with skin cutaneous melanoma (SKCM), stomach adenocarcinoma (STAD), breast cancer (BRCA), lung adenocarcinoma (LUAD), and lung squamous cell carcinoma (LUSC) from 95 distinct clinical sites. These five cancer types were selected given their relevance to immuno-oncology therapies and their image availability in TCGA. We summarize the characteristics of TCGA patients in Supplementary Table 1. To supplement the TCGA analysis cohort, we obtained 4158 additional WSIs for the five cancer types to improve model robustness.

To maximize capture of this information, we excluded images (n = 91, 3.1%) if they failed basic quality control checks as determined by expert pathologists. Criteria for quality control were limited to mislabeling of cancer type, excessive blur, or insufficient staining. For both TCGA and additional WSIs, we collected cell- and tissue-level annotations from a network of pathologists, amounting to >1.4 million cell-type point annotations and >200,000 tissue-type region annotations (Supplementary Table 2).

We used the resulting slides and annotations to design a fully automated pipeline to extract HIFs from these slides (summarized in Fig. 1a). First, we trained deep learning models for cell detection (cell-type models) and tissue region segmentation (tissue-type models). Training and validation of models was conducted on a development set of 1561 TCGA WSIs, supplemented by the 4158 additional WSIs (n = 5719) (Fig. 1b). Next, we exhaustively generated cell- and tissue-type model predictions for 2826 TCGA WSIs, which were then used to compute a diverse array of HIFs for each WSI. Finally, we trained classical linear machine learning models to predict treatment-relevant molecular expression phenotypes using these HIFs.

**a**

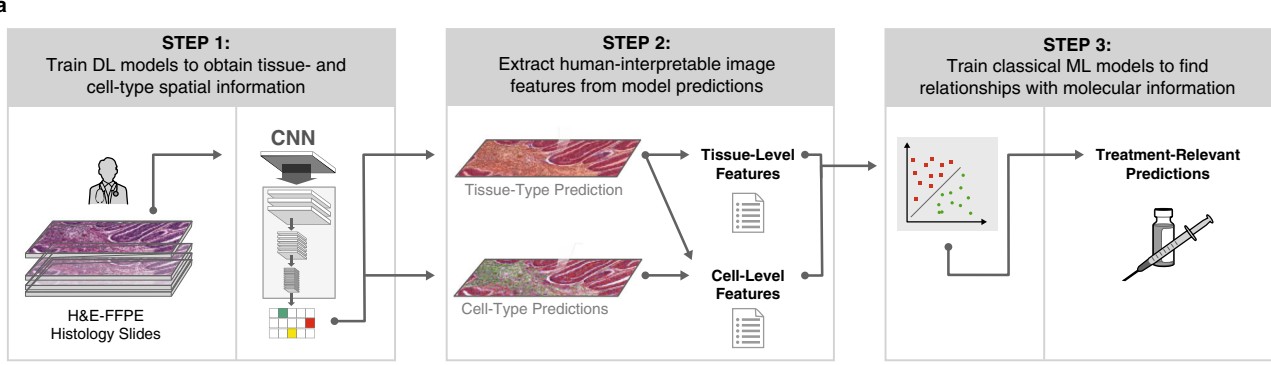

**b**

| TCGA Datasets | BRCA | LUAD | LUSC | SKCM | STAD | Total |
|---|---|---|---|---|---|---|
| **Number of WSIs** | **1119** | **501** | **446** | **358** | **407** | **2831** |
| Number of Distinct Patients | 1044 | 444 | 412 | 327 | 407 | 2634 |
| Number of Cell-Level Annotations | 57840 | 21958 | 68548 | 105096 | 69598 | 323040 |
| Number of Tissue-Level Annotations | 7637 | 5555 | 13826 | 19611 | 8917 | 55546 |
| **Additional Datasets** | | | | | | |
| Number of WSIs | 698 | 1908 | 438 | 1002 | 112 | 4158 |
| Number of Cell-Level Annotations | 130474 | 257966 | 45061 | 658204 | 27999 | 1119704 |
| Number of Tissue-Level Annotations | 23094 | 39195 | 14345 | 61254 | 8130 | 146018 |

**c**

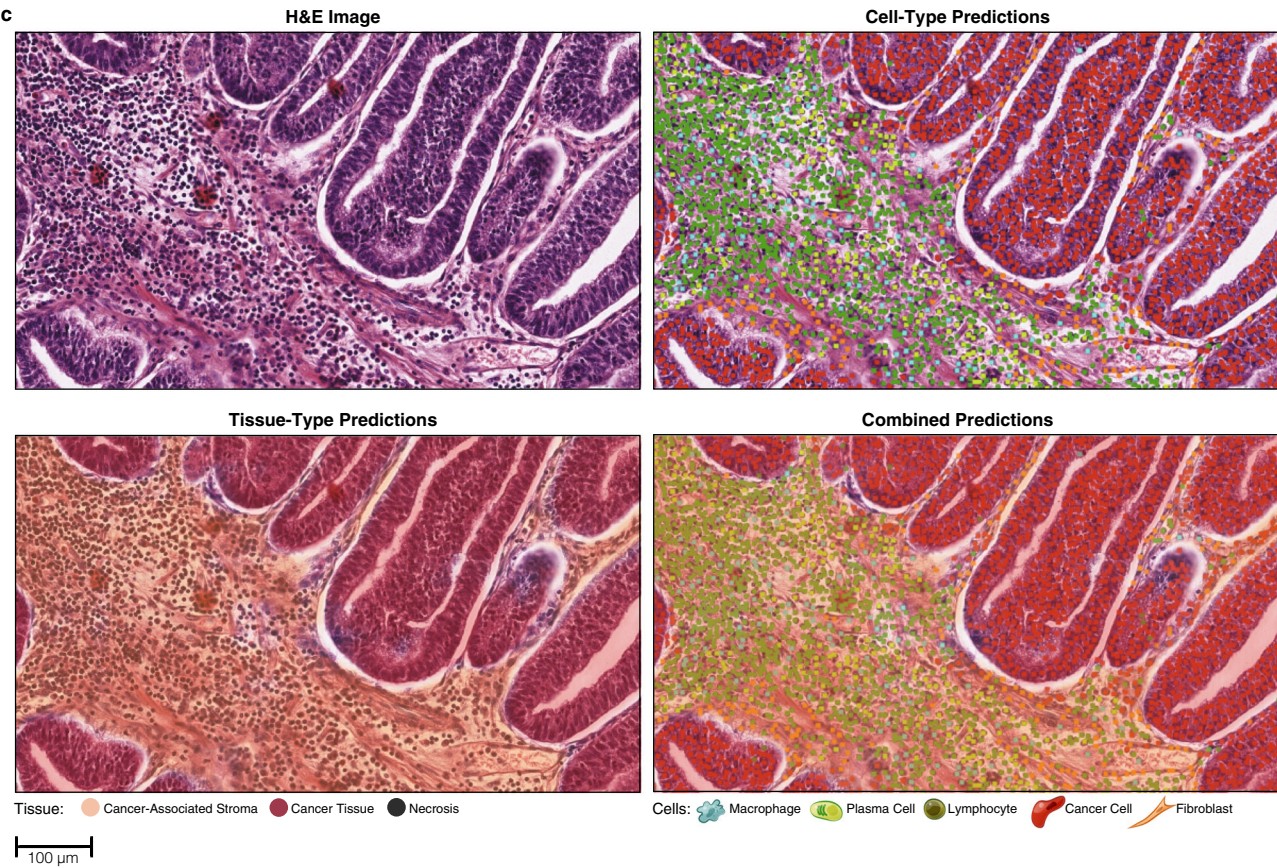

**Fig. 1 Dataset and pipeline overview. a** Methodology for extracting human-interpretable image features (HIFs) from high-resolution, digitized images stained with hematoxylin and eosin (H&E). **b** Summary statistics on the number of whole-slide images (WSIs), distinct patients, and annotations curated from The Cancer Genome Atlas (TCGA) and additional datasets. **c** Unprocessed portions of stomach adenocarcinoma (STAD) H&E-stained slides alongside corresponding heatmap visualizations of cell- and tissue-type predictions. Slide regions are classified into tissue types: cancer tissue (red), cancer-associated stroma (orange), necrosis (black), or normal (transparent). Pixels in cancer tissue or cancer-associated stroma areas are classified into cell types: lymphocyte (green), plasma cell (lime), fibroblast (orange), macrophage (aqua), cancer cell (red), or background (transparent).

**Cell- and tissue-type model development and evaluation.** In the first step of our pipeline, we trained two convolutional neural networks (CNNs) per cancer type: (1) tissue-type models trained to segment cancer tissue, cancer-associated stroma (CAS), and necrotic tissue regions and (2) cell-type models trained to detect lymphocytes, plasma cells, fibroblasts, macrophages, and cancer cells. These models were improved iteratively through a series of quality control steps, including significant input from board-certified pathologists ("Methods"). These CNNs were then used to exhaustively generate cell-type labels and tissue-type segmentations for each WSI. We visualized these predictions as colored heatmaps projected onto the original WSIs (Fig. 1c and Supplementary Fig. 1). Throughout model development, we tracked accuracy metrics on a comprehensively annotated validation dataset (Supplementary Fig. 2).

To directly compare the quality of our cell-type model predictions against pathologist annotation, we generated 250 $75 \times 75$ μm frames of cell-type overlays evenly sampled across the 5 cancer types and 5 cell types, each from a distinct WSI. These frames were then annotated for each of the five cell types by multiple external board-certified pathologists, enabling us to compare cell-type counts as predicted by our CNN cell-type model against pathologist annotation counts. We observed that Pearson correlations between cell-type model predictions and pathologist consensus were comparable to inter-pathologist correlation (differences in correlation ranged from $-0.019$ to 0.024, with a median absolute difference of 0.069) across the five cell types (Supplementary Fig. 3). Model versus pathologist consensus and inter-pathologist correlations were both strong ($>0.8$) for cancer cells and lymphocytes and moderate (approximately 0.4–0.7) for plasma cells, macrophages, and fibroblasts. To assess model generalizability, we redeployed our BRCA cell-type model to predict cell types on H&E, FFPE WSIs from an external BRCA dataset uploaded by Peikari et al. to The Cancer Imaging Archive (TCIA)[35]. We then repeated the same frame analysis framework using 250 frames evenly sampled across the five cell types, which revealed robust concordance between our cell-type model and pathologist consensus in these external WSIs (Supplementary Fig. 4). Correlation coefficients ranged from 0.607 in macrophages to 0.926 in lymphocytes and differed from inter-pathologist correlation by a median absolute difference of 0.076. As a benchmark, inter-pathologist correlation represents the optimal performance that can be expected from models trained and evaluated using pathologist annotations as the ground truth. External data were not publicly available for the remaining cancer types. While the BRCA cell-type model generalized without additional tuning, other models may require retraining when applied to new datasets.

**Cell- and tissue-type predictions yield a wide spectrum of HIFs.** When quantified, our cell- and tissue-type predictions capture broad multivariate information about the spatial distribution of cells and tissues in each slide. Specifically, we used model predictions to extract 607 HIFs (Fig. 2), which can be understood in terms of 6 categories (Fig. 3). The first category includes cell-type counts and densities across different tissue regions (e.g., density of plasma cells in cancer tissue; Fig. 3i, ii). The next category includes cell-level cluster features that capture inter-cellular spatial relationships, such as cluster dispersion, size, and extent (e.g., mean cluster size of fibroblasts in CAS; Fig. 3iii, iv). The third category captures cell-level proportion and proximity features, such as the proportional count of lymphocytes versus fibroblasts within 80 microns (μm) of the cancer–stroma interface (CSI; Fig. 3v, vi). The fourth category includes tissue area (e.g., mm² of necrotic tissue) and multiplicity counts (e.g., number of

significant regions of cancer tissue) (Fig. 3vii, viii). The fifth category includes tissue architecture features, such as the average solidity (solidness) of cancer tissue regions or the fractal dimension (geometrical complexity) of CAS (Fig. 3ix, x). The final category captures tissue-level morphology using metrics such as perimeter[2] over area (shape roughness), lacunarity (gappiness), and eccentricity (Fig. 3xi, xii). This broad enumeration of biologically relevant HIFs explores a wide range of mechanisms underlying histopathology across diverse cancer types.

**HIFs capture sufficient information to stratify cancer types.** To visualize the global structure of the HIF feature matrix, we used Uniform Manifold Approximation and Projection (UMAP)[36,37] to reduce the 607-dimensional HIF space into two dimensions (Fig. 4a). The two-dimensional (2-D) manifold projection of HIFs was able to separate BRCA, SKCM, and STAD into distinct clusters, while merging NSCLC subtypes LUAD and LUSC into one overlapping cluster (V-measure score = 0.47 using $k$-means with $k = 4$).

Cancer-type differences could be traced to specific and interpretable cell- and tissue-level features within the TME (Fig. 4b). SKCM samples exhibited higher densities of cancer cells in CAS (pan-cancer median $Z$-score = 0.55, $P < 10^{-30}$) and greater cancer tissue area per slide ($Z$-score = 0.72, $P < 10^{-30}$) relative to other cancer types. These findings reflect biopsy protocols for SKCM, in which the excised region involves predominantly cancer tissue and minimal normal tissue. NSCLC subtypes LUAD and LUSC exhibited higher densities of macrophages in CAS ($Z$-score = 0.54 and 0.91, respectively; $P < 10^{-30}$), reflecting the large population of macrophages infiltrating alveolar and interstitial compartments during lung inflammation[38]. NSCLC subtypes also exhibited higher densities of plasma cells ($Z$-score = 0.61 and 0.49; $P < 10^{-30}$) in CAS, in agreement with prior findings in which proliferating B cells were observed in ~35% of lung cancers[39,40]. STAD exhibited the highest density of lymphocytes in CAS ($Z$-score = 0.11, $P = 2.16 \times 10^{-19}$), corroborating prior work that identified STAD as having the largest fraction of TIL-positive patches per WSI among 13 TCGA cancer types, including the 5 examined here[30]. Notably, HIFs are able to stratify cancer types by known histological differences without explicit tuning for cancer-type detection, as is required by "black box" approaches. In a stratified analysis, SKCM metastatic and primary tumor samples also exhibited notable differences, including a greater average solidity and area of cancer tissue among metastatic tumors (Supplementary Fig. 5). Considering spatial heterogeneity, we observed an enrichment of lymphocytes and plasma cells in SKCM as well as an enrichment of cancer cells in LUSC and LUAD at the CSI relative to in cancer tissue plus CAS (CT + CAS) (Supplementary Fig. 6).

**HIFs are concordant with sequencing-based cell and immune signature quantifications.** To further validate our deep learning-based cell quantifications, we compared the abundance of the same cell type predicted by our cell-type models with those based on RNA sequencing (RNA-Seq)[41]. Image-based cell quantifications were correlated with sequencing-based quantifications across all patient samples and cancer types (pan-cancer) in three cell types (Supplementary Fig. 7): leukocyte fraction (Spearman correlation coefficient ($\rho$) = 0.55, $P < 2.2 \times 10^{-16}$), lymphocyte fraction ($\rho$ = 0.42, $P < 2.2 \times 10^{-16}$), and plasma cell fraction ($\rho$ = 0.40, $P < 2.2 \times 10^{-16}$). Notably, imperfect correlation is expected as tissue samples used for RNA-Seq and histology imaging are

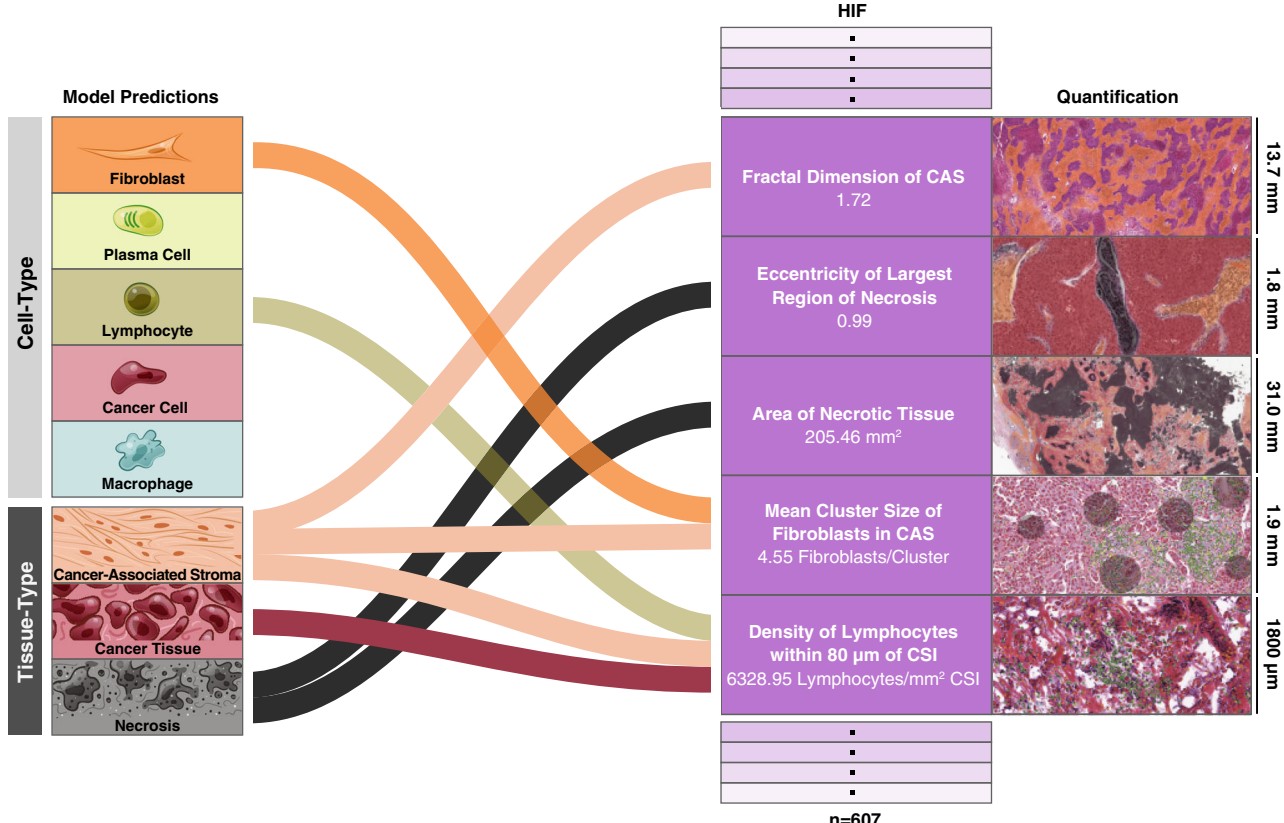

**Fig. 2 Human-interpretable image feature extraction workflow.** Flow diagram for extraction of human-interpretable image features (HIFs) from model predictions for five example HIFs. For each HIF, a histological snapshot with the corresponding cell- or tissue-type heatmap overlaid and the associated quantity are shown. Histological snapshots have dimensions (from top to bottom, width by height) of 24,000 × 13,693, 3126 × 1785, 54,303 × 30,981, 3422 × 1953, and 3110 × 1800 µm. Cell- and tissue-type color coding is the same as Fig. 1c.

extracted from different portions of the patient's tumor and thus vary in TME due to spatial heterogeneity.

There is significant correlation structure among individual HIFs due to the modular process by which feature sets are generated, as well as inherent correlations in underlying biological phenomena. For example, proportion, density, and spatial features of a given cell or tissue type all rely on the same underlying model predictions. In order to identify mechanistically relevant and inter-correlated groups of HIFs, hierarchical agglomerative clustering was conducted ("Methods"; Supplementary Data 1). This clustering also increases the power of multiple-hypothesis-testing corrections by accounting for feature correlation[42]. Pan-cancer HIF clusters strongly correlated with immune signatures of leukocyte infiltration, immunoglobulin G (IgG) expression, transforming growth factor (TGF)-β expression, and wound healing (Fig. 5a), as well as angiogenesis and hypoxia (Supplementary Fig. 8), all quantified by scoring bulk RNA-Seq reads for known immune and gene expression signatures[43–45]. We conducted the same correlational analysis for each cancer type individually and observed high concordance among the top correlated HIF clusters per immune signature (Supplementary Table 3).

Molecular quantification of leukocyte infiltration was concordant with the density of leukocyte-lineage cells in CT + CAS quantified by our deep learning pipeline, including lymphocytes (median absolute Spearman correlation $\rho$ for associated HIF cluster = 0.48, $P < 10^{-30}$; Fig. 5bi), plasma cells (cluster $\rho = 0.46$, $P < 10^{-30}$), and macrophages (cluster $\rho = 0.40$, $P < 10^{-30}$). Similarly, we observed associations between IgG expression and the density of leukocyte-lineage cells in CT + CAS, with

plasma cells being the most strongly correlated (cluster $\rho = 0.58$, $P < 10^{-30}$), as expected given their role in producing Igs (Fig. 5bii). TGF-β expression was associated with the density of fibroblasts in CT + CAS (cluster $\rho = 0.28$, $P < 10^{-30}$; Fig. 5biii), building upon prior studies which found that TGF-β1 can promote fibroblast proliferation[46–48]. Interestingly, recent studies in breast and ovarian cancer have highlighted the role of several subsets of cancer-associated fibroblasts in promoting an immunosuppressive environment resistant to anti-programmed cell death protein 1 (anti-PD-1) therapy, including one subset associated with the TGF-β signaling pathway[49]. TGF-β expression was also correlated with the area of CAS relative to CT + CAS (cluster $\rho = 0.31$, $P < 10^{-30}$), shedding further light on the role of stromal proteins in modulating TGF-β levels[50]. The wound healing signature was positively associated with the density of fibroblasts in CAS versus in cancer tissue (cluster $\rho = 0.29$, $P < 10^{-30}$; Fig. 5biv), which corroborates findings that both tumors and healing wounds alike modulate fibroblast recruitment and proliferation to facilitate extracellular matrix deposition[51]. H&E snapshots corresponding to high expression of each of the four immune signatures are shown in Fig. 5c with corresponding cell-type heatmaps overlaid.

The angiogenesis signature was positively associated with the density of fibroblasts (cluster $\rho = 0.32$, $P < 10^{-30}$) and macrophages (cluster $\rho = 0.31$, $P < 10^{-30}$) in CAS, corroborating the critical role that fibroblasts and macrophages play in modulating extracellular matrix components to promote neovascularization[52,53]. Interestingly, angiogenesis signature was also associated with the area of CAS relative to CT + CAS (cluster $\rho = 0.29$, $P < 10^{-30}$), reflecting the importance of stromal cell populations (Supplementary Fig. 8).

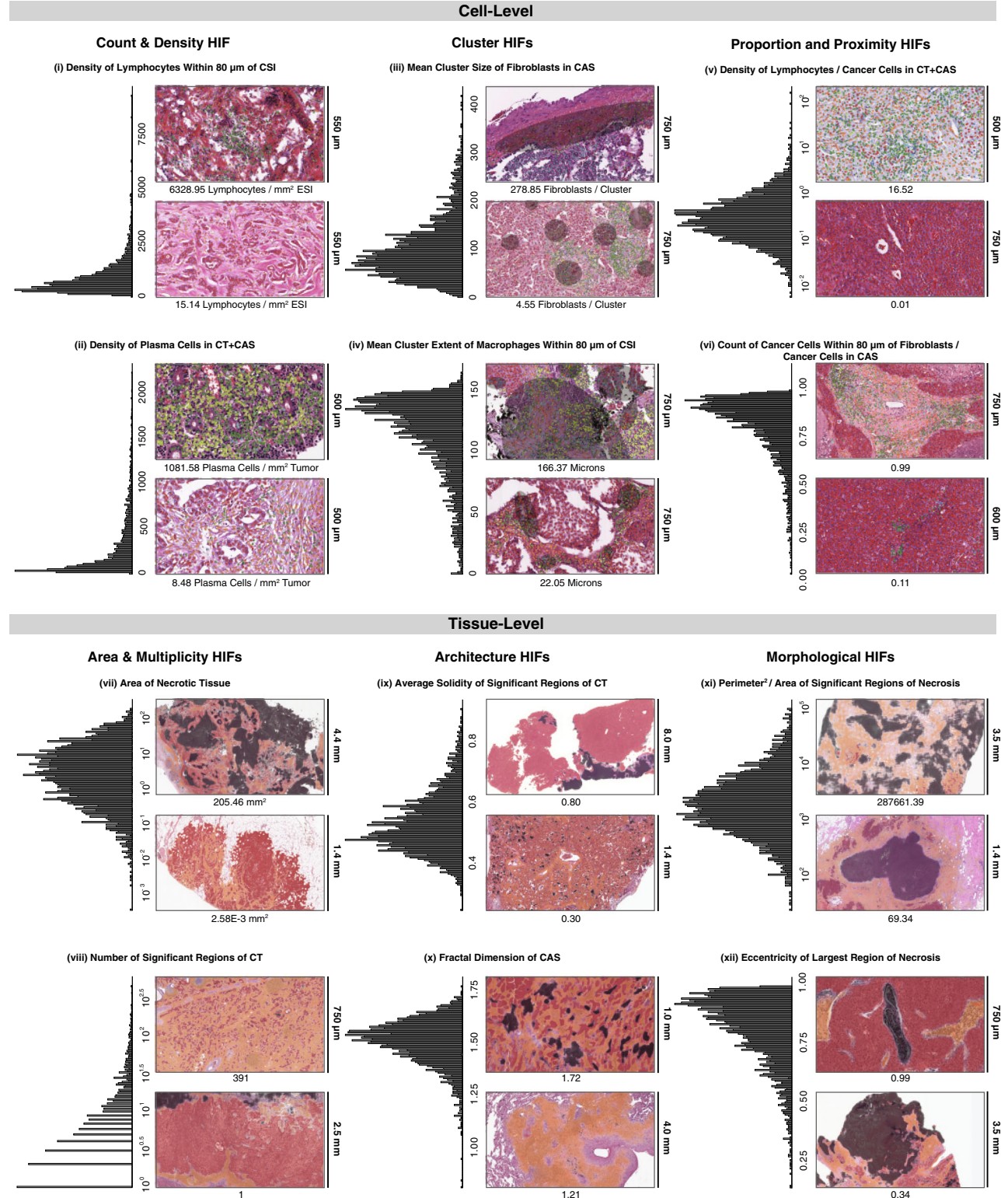

The hypoxia signature was most strongly associated with area of necrotic tissue (cluster $\rho = 0.45$, $P < 10^{-30}$), as expected by their causal relationship (Supplementary Fig. 8). Hypoxia was also associated with density of plasma cells in CAS (cluster $\rho = 0.36$, $P < 10^{-30}$), which confirms prior findings of increased plasma cell generation under hypoxic conditions[54].

While many associations noted above have been previously identified using experimental methods, a HIF-based approach

enables validation and systematic quantification of the strength of such associations.

**HIFs are predictive of clinically relevant phenotypes.** To evaluate the capability of HIFs to predict expression of clinically relevant, immuno-modulatory genes, we conducted supervised prediction of binarized classes for five clinically relevant phenotypes: (1) PD-1 expression, (2) PD-L1 expression, (3) cytotoxic

**Fig. 3 Overview of HIFs.** Graphical overview of the 607 human-interpretable image features (HIFs) grouped into six categories: cell-level count and density ($n = 56$ HIFs), cell-level cluster ($n = 180$), cell-level proportion and proximity ($n = 208$), tissue-level area and multiplicity ($n = 13$), tissue-level architecture ($n = 25$), and tissue-level morphology ($n = 125$). For each HIF, a histogram of the HIF quantified in all patient samples across the five cancer types and histological snapshots corresponding to high and low values with the corresponding heatmap are shown. Both snapshots are taken from patient samples of the same cancer type. Cell- and tissue-type heatmaps adhere to the same color scheme described in Fig. 1c. In (iii), fibroblast clusters are annotated, contrasting one large cluster against multiple smaller clusters. In (iv), macrophage clusters and extents are annotated. Cluster extent is defined as the maximum distance between a cluster exemplar (defined via Birch clustering) and a cell within that cluster. Significant regions (viii) are defined as connected components (identified at the pixel level) of a given tissue type with at least 10% the size of the largest connected component in the slide. A solidity (ix) of one corresponds to a completely filled object, while values less than one correspond to objects containing holes or with irregular boundaries. Fractal dimension (x) can efficiently estimate the geometrical complexity and irregularity of shapes and patterns, thus capturing tissue architecture. A fractal dimension of one corresponds to a perfectly smooth tissue border, while higher fractal dimension corresponds to increasing roughness and irregularity, indicating more extensive physical contact between adjacent tissue types. The fractal dimension of the cancer–stroma interface (CSI) has been previously associated with dysfunction in antigen presentation[29]. Perimeter²/area (xi) is a unitless measure of shape roughness (e.g., square $= 16$, circle $= 4\pi$). Across all HIFs, tumor regions include cancer tissue (CT), cancer-associated stroma (CAS), and a combined CT + CAS. Cell- and tissue-type color coding is the same as Fig. 1c.

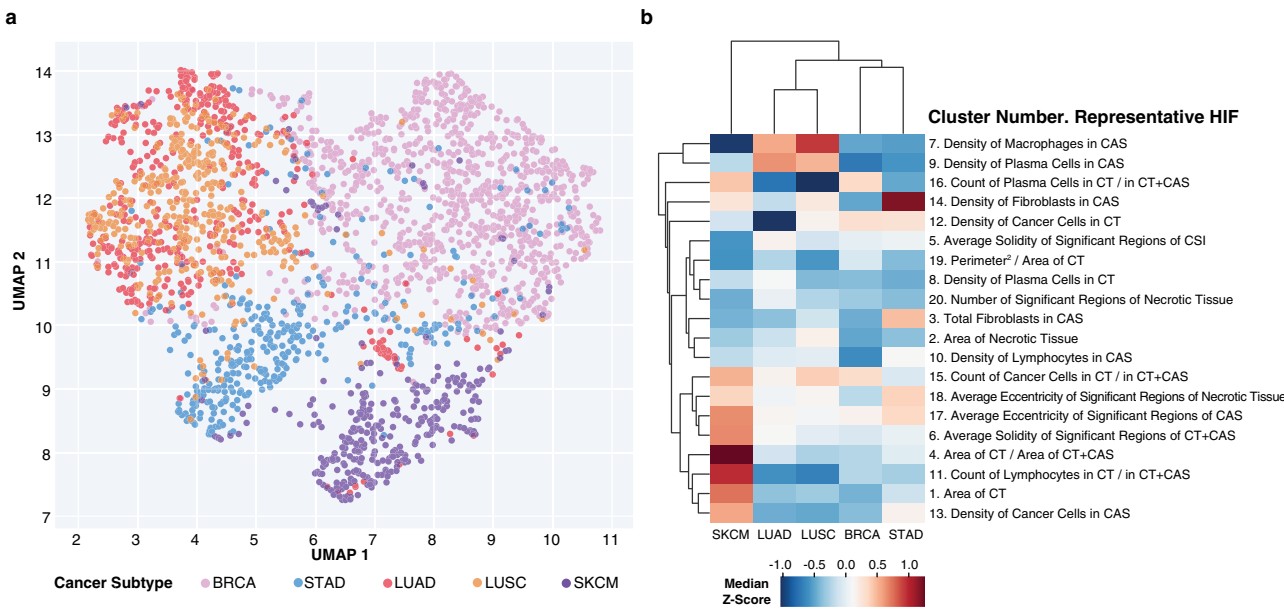

**Fig. 4 HIF differences across cancer types. a** Uniform Manifold Approximation and Projection (UMAP) visualization of five cancer types reduced from the 607-dimension space defined by human-interpretable image feature (HIF) values into two dimensions. Each point represents a patient sample colored by cancer type. **b** Clustered heatmap of median Z-scores (computed pan-cancer) across cancer types for 20 HIFs, each representing one HIF cluster (defined pan-cancer). Hierarchical clustering was performed using average linkage and Euclidean distance. Clusters are annotated with a representative HIF chosen based on interpretability and high variance across cancer types.

T-lymphocyte-associated protein 4 (CTLA-4) expression, (4) homologous recombination deficiency (HRD) score, and (5) T cell immunoreceptor with Ig and ITIM domains (TIGIT) expression (Fig. 6 and Supplementary Fig. 9). Using the 607 HIFs computed per WSI, predictions were conducted for cancer types individually as well as pan-cancer. SKCM predictions were conducted only for TIGIT expression due to insufficient sample sizes for the remainder of outcomes ("Methods"). To demonstrate model generalizability across varying patient demographics and sample collection processes, area under the receiver operating characteristic (AUROC) and area under the precision-recall curve (AUPRC) performance metrics were computed on hold-out sets composed exclusively of patient samples derived from tissue source sites not seen in the training sets (Supplementary Table 4).

HIF-based models were not predictive for every phenotype in each cancer type (hold-out AUROC < 0.6; see Supplementary Table 5 for all results including negatives). In the successful prediction models (hold-out AUROC range = 0.601–0.864;

Fig. 6a), precision-recall curves revealed that models were robust to class imbalance, achieving AUPRC performance surpassing positive class prevalence by 0.104–0.306 (Supplementary Fig. 10).

On average across molecular phenotype prediction tasks, AUROC hold-out performance of our HIF-based linear models was comparable to that achieved by end-to-end deep learning models trained using the same architecture and hyper-parameters from Kather et al. (Supplementary Table 6)[11]. Differences in AUROC ranged from −0.16 to 0.25, with a median absolute difference of 0.065. Given the small sample sizes, HIF-based models are potentially better statistically powered. Indeed, HIF-based models outperformed end-to-end models in several prediction tasks, including most notably SKCM prediction of TIGIT expression, which boasted the smallest sample size. AUROC performance of our HIF-based linear model for PD-L1 expression in LUAD trained on roughly 300 WSIs was also comparable to that achieved by previously published "black-box" deep learning models trained on hundreds of thousands of paired H&E and PD-L1 example patches in NSCLC[55].

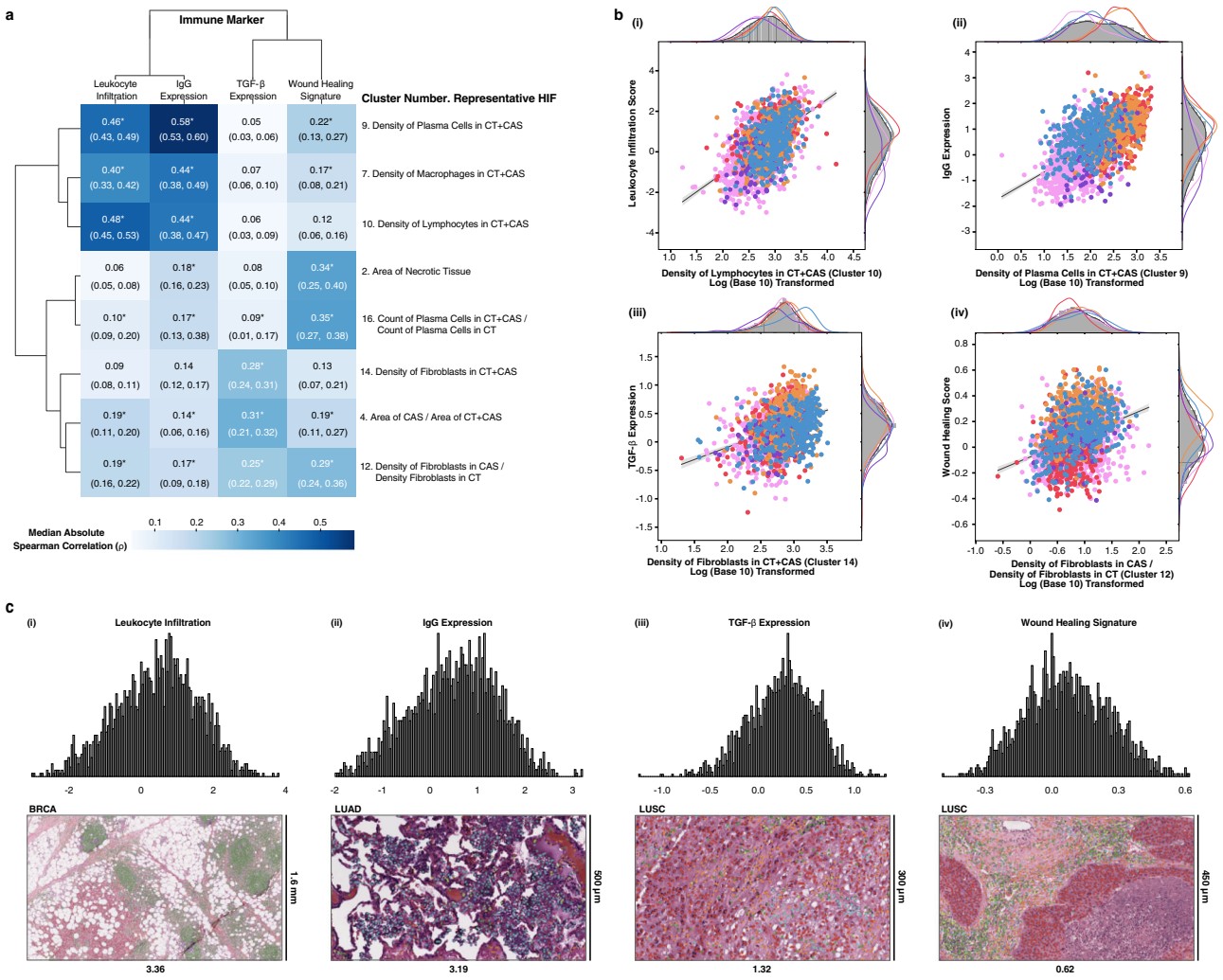

**Fig. 5 Validation of HIFs against immune signatures. a** Clustered heatmap of median absolute Spearman correlation coefficients ($\rho$) computed across all patient samples between eight clusters of human-interpretable image features (HIFs) (defined pan-cancer) and four canonical immune signatures. *P* values were computed using a two-sided test for whether the correlation coefficient was significantly different from 0. Hierarchical clustering was done using average linkage and Euclidean distance. Median absolute Spearman correlation coefficients with a combined (via the Empirical Brown's method) and corrected (via the Benjamini–Hochberg procedure) *P* value lower than the machine precision level ($10^{-30}$) are annotated with an asterisk. Negative control analyses are included in Supplementary Table 3. Tumor regions include cancer tissue (CT), cancer-associated stroma (CAS), and a combined CT + CAS. **b** Correlation and kernel density estimation plots between representative HIFs and immune signatures. Points are colored by cancer type (same schema as Fig. 4a). *X*-axes are log-transformed (base ten). Trendlines are plotted on the log-transformed data. Cell densities are reported in count/mm$^2$ and tissue areas are reported in mm$^2$. **c** Histogram of immune signature expression (*Z*-score) across all patients, alongside a histological snapshot with its cell-type heatmap overlaid corresponding to high expression of the given immune signature. Cell-type heatmaps adhere to the same color scheme described in Fig. 1c. Histological snapshots have dimensions (from left to right, width by height) of 12,479 × 7109, 4230 × 2408, 4230 × 2408, and 5286 × 3016 μm.

While our HIF generation process explicitly encodes for interactions between biological entities (e.g., count of lymphocytes within 80 μm of fibroblasts), we also compared and achieved comparable hold-out AUROC and AUPRC performance between our HIF-based linear models against HIF-based random forest models, which directly account for interaction effects between HIFs (Supplementary Table 7).

**Predictive HIFs provide interpretable link to clinically relevant phenotypes.** Interpretable features enable interrogation and further validation of model parameters as well as generation of biological hypotheses. Toward this end, for each prediction task we identified the five most important HIF clusters as determined by magnitude of model coefficients (Fig. 6b and Supplementary

Fig. 11) and computed cluster-level *P* values to evaluate significance (Supplementary Table 8; "Methods").

As expected, prediction of PD-1 and PD-L1 involved similar HIF clusters (Pearson correlation between PD-1 and PD-L1 expression = 0.53; Supplementary Fig. 12). For example, the extent of tumor inflammation, as measured by the count of cancer cells within 80 μm of lymphocytes, as well as the density of lymphocytes in CT + CAS, was significantly selected during model fitting for both of PD-1 and PD-L1 expression in pan-cancer and BRCA models (Fig. 6bi, ii and Supplementary Fig. 11i, ii). Furthermore, in both LUAD and LUSC, the count of lymphocytes in CT + CAS was similarly predictive of PD-1 and PD-L1 expression. The importance of these HIFs that capture lymphocyte infiltration between and surrounding cancer cells corroborates prior literature, which demonstrated that TILs

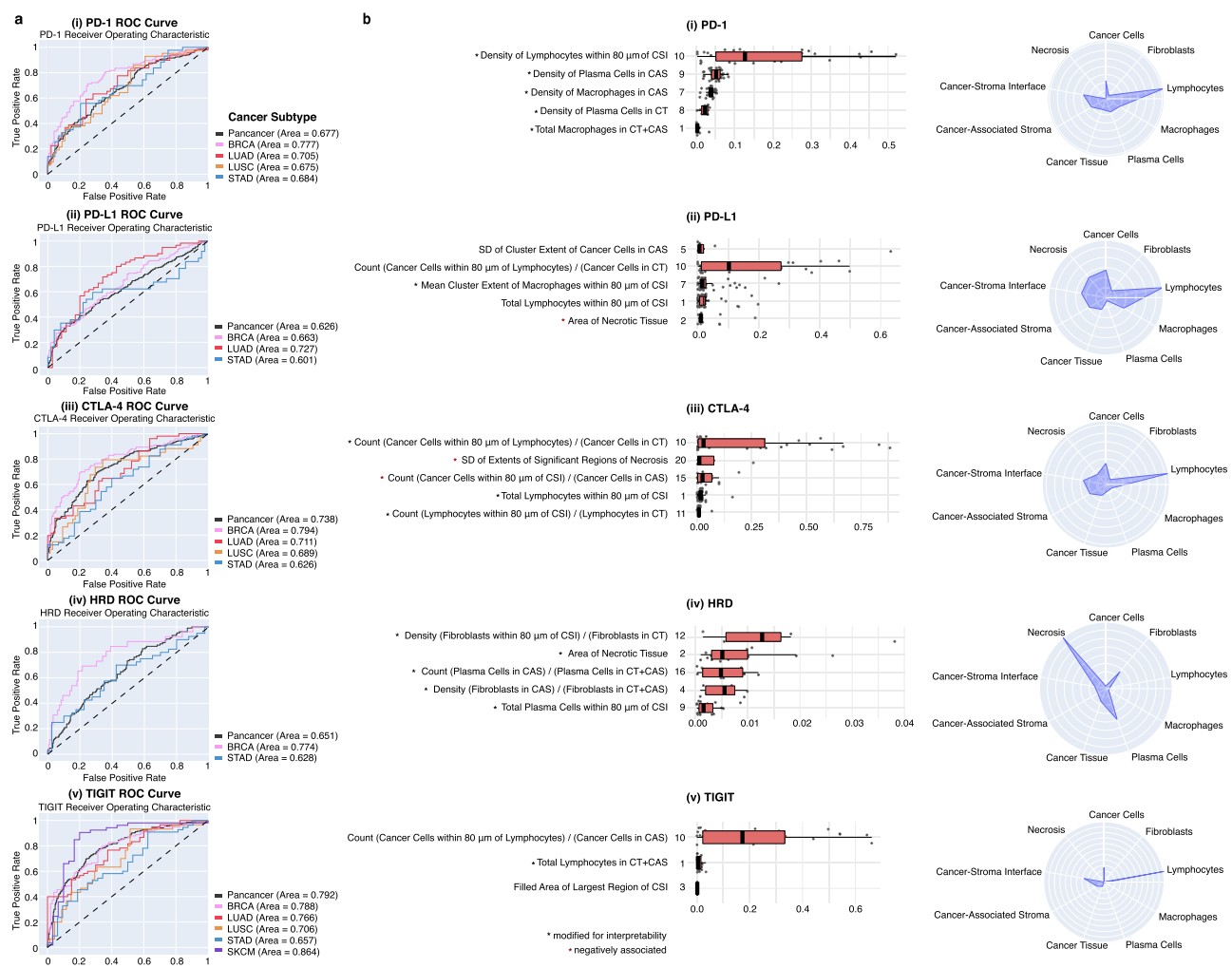

**Fig. 6 HIF-based prediction of molecular phenotypes. a** Receiver operator characteristic (ROC) curves for (i) PD-1, (ii) PD-L1, (iii) CTLA-4, (iv) HRD, and (v) TIGIT hold-out predictions across cancer types and pan-cancer. Skin cutaneous melanoma (SKCM) predictions were conducted only for TIGIT due to low sample sizes. Pan-cancer predictions use binary labels thresholded independently by cancer type. For TIGIT predictions, pan-cancer includes all five cancer types. For the remainder of predictions, pan-cancer includes all cancer types excluding SKCM. Random classifiers correspond to area under the ROC curve (AUROC) = 0.50. **b** Visualization of predictive human-interpretable image features (HIFs) for each molecular phenotype. Boxplots show the top five most predictive HIF clusters for each phenotype in pan-cancer models. For TIGIT predictions, pan-cancer models only included three non-zero HIF clusters. Clusters are ranked by the maximum absolute ensemble beta across HIFs in a given cluster. Ensemble betas are computed per HIF as the average across the three models incorporated into the final ensemble evaluated on the hold-out set. The center and bounds of each boxplot represent the median and interquartile range (IQR; 25th, 75th percentiles) for HIF betas in each cluster, respectively. Upper and lower boxplot whiskers represent the smaller of the maximum beta value or the 75th percentile + 1.5 × IQR and the larger of the minimum beta value or the 25th percentile − 1.5 × IQR, respectively. Each cluster is labeled with a representative HIF corresponding to the maximum absolute ensemble beta value. The number of ensemble betas (HIFs) used to derive each boxplot is: 32, 49, 32, 9, and 11 (from top to bottom) for PD-1 clusters; 8, 30, 49, 20, and 70 for PD-L1 clusters; 38, 4, 14, 77, and 20 for CTLA-4 clusters; 7, 15, 11, 8, and 19 for HRD clusters; and 26, 22, and 2 for TIGIT clusters (see Supplementary Data 1 for the number of HIFs per cluster). In cases where that HIF is difficult to interpret, a more interpretable HIF within a fivefold difference of the maximum ensemble beta is presented (indicated by a black asterisk). As absolute values were used for ranking, HIFs with negative ensemble betas are denoted by a red asterisk. Boxplots of predictive HIF clusters for cancer type-specific models are included in Supplementary Fig. 11. Radar charts show the normalized magnitude of ensemble betas in pan-cancer models stratified across nine HIF axes, corresponding to the five cell types, three tissue types, and cancer–stroma interface (CSI). Normalized magnitudes were computed as the sum of absolute ensemble betas for HIFs associated with each axis divided by the total number of HIFs associated with the said axis (e.g., all HIFs involving fibroblasts). Multiple predictive HIFs are visualized with overlaid cell- or tissue-type heatmaps in Fig. 3. Tumor regions include cancer tissue (CT), cancer-associated stroma (CAS), and a combined CT + CAS.

correlated strongly with higher expression levels of PD-1 and PD-L1 in early BRCA[56] and NSCLC[57,58].

The area, morphology, or multiplicity of necrotic tissue proved predictive of PD-1 expression in LUAD, LUSC, and STAD models and of PD-L1 expression in pan-cancer, BRCA, and LUAD models, expanding upon prior findings that tumor necrosis correlated positively with PD-1 and PD-L1 expression in LUAD[59]. The density, proximity, or clustering properties of plasma cells was predictive of PD-1 expression in all models excluding LUAD, suggesting a role for plasma cells in modulating PD-1 expression. Recent studies in SKCM, renal cell carcinoma, and soft-tissue sarcoma have demonstrated that an enrichment of B cells in tertiary lymphoid structures was positively predictive of response to immune checkpoint blockade therapy[60–62]. The

density of fibroblasts in CAS or within 80 µm of the CSI was predictive of PD-L1 expression in LUAD and STAD, respectively, corroborating earlier discoveries that cancer-associated fibroblasts promote PD-L1 expression[63].

Less is known about the relationship between the TME and CTLA-4 expression. By investigating predictive HIFs, we can begin to enumerate features of the TME that correlate with CTLA-4 expression. The proximity of lymphocytes to cancer cells (pan-cancer and BRCA), morphology of necrotic regions (LUAD and LUSC), and density of cancer cells in CT + CAS versus exclusively in CAS (BRCA and STAD) were predictive of CTLA-4 expression across multiple models (Fig. 6biii and Supplementary Fig. 11iii).

Area of necrotic tissue (pan-cancer and BRCA) as well as various morphological properties of necrotic regions including perimeter and lacunarity (BRCA and STAD) was predictive of HRD (Fig. 6biv and Supplementary Fig. 11iv). In HRD, ineffective DNA damage repair can result in the accumulation of severe DNA damage and subsequent cell death through apoptosis as well as necrosis[64,65]. The density and count of fibroblasts near or in CAS was also predictive of HRD in the pan-cancer and BRCA models, corroborating prior findings that persistent DNA damage and subsequent accumulation of unrepaired DNA strand breaks can induce reprogramming of normal fibroblasts into cancer-associated fibroblasts[66].

Like the three other immune checkpoint proteins (PD-1, PD-L1, and CTLA-4), TIGIT expression was also associated with markers of tumor inflammation, including the count of cancer cells within 80 µm of lymphocytes (pan-cancer and BRCA), the total number of lymphocytes in CT + CAS (pan-cancer and BRCA), and the proportional count of lymphocytes to cancer cells within 80 µm of the CSI (LUAD) (Fig. 6bv and Supplementary Fig. 11v). These findings corroborate prior findings that TIGIT expression, alongside PD-1 and PD-L1 expression (Pearson correlation between TIGIT and PD-1 = 0.84; TIGIT and PD-L1 = 0.56; Supplementary Fig. 12), is correlated with TILs[67]. HIF clusters capturing morphology and architecture of necrotic tissue (e.g., fractal dimension, lacunarity, extent, perimeter$^2$/area) were associated with TIGIT expression in LUAD, LUSC, SKCM, and STAD models, although these relationships have yet to be investigated.

## Discussion

In recent years, fusion approaches that combine deep learning with feature engineering have gained traction[68–71]. Our study combines exhaustive deep learning-based cell- and tissue-type classifications to compute image features that are both biologically relevant and human interpretable. We demonstrate that computed HIFs can recapitulate sequencing-based cell quantifications, capture canonical immune signatures such as leukocyte infiltration and TGF-β expression, and robustly predict five molecular phenotypes relevant to the efficacy of targeted cancer therapies. We also demonstrate the generalizability of our associations, as evidenced by similarly predictive HIF clusters across biopsy images derived from five different cancer types. Notably, we show that our HIF-based approach, which integrates the predictive power of deep learning with the interpretability of feature engineering, achieves comparable performance to that of black-box models.

While prior studies have applied deep learning methodologies to capture cell-level information, such as the spatial configuration of immune and stromal cells[29,71], or tissue-level information[72] alone, our combined cell and tissue approach enables quantification of increasingly complex and expressive features of the TME, ranging from the mean cluster size of fibroblasts in CAS to

the proximity of TILs or cancer-associated fibroblasts to the CSI. For instance, while TILs are emerging as a promising biomarker in solid tumors such as triple-negative and HER2-positive breast cancer[71], TILs differ from stromal lymphocytes, and substantial signal can be obtained by considering multiple cell–tissue combinations[25]. By training models to make six-class cell-type and four-class tissue-type classifications from >1.6 million pathologist annotations, our approach is also able to capture more interactions between cell types and tissue regions than prior HIF-based studies[25–30].

Our approach exhaustively generates cell- and tissue-type predictions across entire WSIs at subcellular resolution (2 and 4 µm, respectively) and improves upon previous tiling approaches that downsample the image. The tissue visible in a WSI is already only a fraction of the tumor; using the entire slide reduces the probability of fixating on local effects and enables quantification of complex characteristics that span multiple tissue regions (e.g., multiplicity, solidity, and fractal dimension of necrotic regions).

In addition, our approach of systematically quantifying specific and interpretable features of the tumor and its surroundings can enable hypothesis generation and a deeper understanding of the TME's influence on drug response. Recent studies provide evidence that the tumor immune architecture may influence the clinical efficacy of immune checkpoint inhibitor[73] and poly (ADP-ribose) polymerase inhibitor therapies[74].

Lastly, during both model development and evaluation, we sought to emphasize robustness to real-world variability[75]. In particular, we supplemented TCGA WSIs with additional diverse datasets during CNN training, integrated pathologist feedback into model iterations, and evaluated HIF-based model performance on hold-out sets composed exclusively of samples from unseen tissue source sites, improving upon prior approaches to predicting molecular outcomes from TCGA H&E images[26,76].

Our study data from TCGA carries several limitations. First, biopsy images submitted to the TCGA dataset are biased toward primary tumors and tumors with more definitive diagnoses that may not generalize well to ordinary clinical settings. Indeed, associations identified in primary tumors may not necessarily generalize to metastatic settings (Supplementary Fig. 5). Second, TCGA is limited to images of H&E staining, which limits the breadth of information available to models. Integrating multi-modal data containing stains against Ki-67 or immunohistological targets may increase confidence in cell classifications[77]. Third, batch effects in TCGA can originate from differing tissue collection, sectioning, and processing procedures. Our validation procedure of partitioning by tissue source site does not account for all possible data artifacts, but it does control for confounding by sample collection, extraction, and other site-specific variables. Our HIF-based approach also limits the impact of spurious associations introduced by batch effects by pre-defining features based on biological phenomena. Fourth, TCGA has limited treatment data and clinical endpoint data are less reliable than molecular data. As TCGA samples were made available in 2013[78], treatment regimens for these cases also predate the widespread adoption of immune checkpoint inhibitors. As such, our models were restricted to prediction of molecular phenotypes with relevance to drug response, in lieu of more direct clinical endpoints, such as RECIST[79] and overall survival. While molecular phenotypes such as PD-L1 expression are informative for clinical endpoints such as sensitivity to immune checkpoint blockade[80], the ability to robustly predict biomarkers does not necessarily translate into robust prediction of relevant endpoints. Ultimately, direct prediction of patient outcomes is needed for clinical integration. Our study provides an interpretable framework to generate hypotheses for clinically relevant biomarkers that can be validated in future prospective studies[81]. The curation of public

datasets with matched pathology images and high-fidelity treatment information could help bridge the remaining gap.

The HIF-based approach also has limitations. First, annotations vary in reliability. Macrophages are particularly difficult for pathologists to identify solely under H&E staining. While the accuracy of an individual pathologist identifying macrophages may be poor, our models represent an aggregate estimate based on training from hundreds of pathologist annotators, which may carry a more reliable signal[82,83]. Future development of our approach could extend to multiplex immunofluorescence technologies that measure spatial protein expression. These methods face challenges of increased cost, lower resolution, and lower scalability across WSIs but may improve upon traditional immunohistochemistry staining in predicting drug response to immune checkpoint inhibitors[84] and reduce the need for expert annotation of cell types. Second, curation of high-fidelity, large-scale pathologist annotations can be time-consuming and expensive. Improvement of open-source segmentation models could accelerate the adoption of HIF-based models. Third, morphologically similar cells (e.g., macrophages, dendritic cells, endothelial cells, pericytes, myeloid-derived suppressor cells, and atypical lymphocytes) may all be captured under a single cell-type prediction. Thus HIFs may, in reality, capture information about a mixture of cell types. For example, in diffuse forms of STAD in which cancer cells invade smooth muscle tissue, our models misclassified certain smooth muscle cells as fibroblasts. Collecting targeted annotations of morphologically similar cell types may decrease noise in HIF estimates and improve performance. Lastly, HIFs are computed as summary statistics within each tissue type across WSIs. Applying "attention-based" HIF computation to focus on regions of interest and further account for spatial heterogeneity is a potential avenue for further research.

Recent work[18–20] has revealed the weaknesses of low-interpretability models, including brittleness to population differences, vulnerabilities to technical artifacts, and susceptibility to unforeseen real-world failure modes. Although HIF-based approaches are not immune to such risks, they provide easier debugging and identification of failure modes than end-to-end models. Beyond suggesting interpretable hypotheses for causal mechanisms (e.g., the anti-tumor effect of high lymphocyte density), our HIF-based approach can be continually validated at several points: pathologists can judge the quality of cell- and tissue-type predictions, estimate the values of each relevant feature using traditional manual scoring, and note when variability in sample preparation or quality may significantly affect relevant features.

Interpretable sets of HIFs, computed from tens of thousands of deep learning-based cell- and tissue-type predictions per patient, improve upon conventional "black-box" approaches that apply deep learning directly to WSIs, yielding models with millions of parameters and limited interpretability. While gradient-based saliency and class activation maps can identify relevant image regions in end-to-end CNN models[11–13], they only enable subjective generation of hypotheses based on slide-by-slide qualitative assessment and are susceptible to human biases[85]. Other model-agnostic interpretability methods, such as partial dependence plots and feature importance measures, are also unable to objectively and scalably connect pixel intensity features to biological phenomena. By contrast, predictive HIFs are directly mapped onto biological concepts and can be interpreted quantitatively across thousands of images. This allows investigators to directly identify concrete hypotheses and correlations that can be investigated further in causal analyses.

Unlike "black-box" models that may opaquely rely on features that are predictive but disconnected from the outcome of interest, such as tissue excision or preparation artifacts (e.g., surgical or

pathologist markings)[20,23], HIF-based predictions can be traced to observable features, allowing model failures to be observed, explained, and addressed. Furthermore, HIF-based models enable users to explicitly define the set of features or hypotheses under examination, reducing the risk of spurious correlations and potentially increasing performance for low sample size prediction tasks. While additional comparative studies are needed, improved trust and reliability against unexpected failures would make HIF-based models a valuable alternative to end-to-end models.

The ability to predict molecular phenotypes directly from WSIs in an interpretable fashion offers numerous potential benefits for clinical oncology. Hospitals, healthcare institutions, and biotechnology companies have decades of archival histopathology data captured from routine care and clinical trials[86]. With improved accuracy, HIF-based models could leverage this information to enable the discovery of patient subpopulations with specific treatment susceptibilities, biomarkers predictive of drug response, and hypotheses for subsequent research.

## Methods

**Dense, high-resolution prediction of cell and tissue types using CNNs**. In order to compute histopathological image features for each slide, it was necessary to first generate cell and tissue predictions per WSI. To this end, we asked a network of board-certified pathologists to label WSIs with both polygonal region annotations based on tissue type (cancer tissue, CAS, necrotic tissue, and normal tissue or background) and point annotations based on cell type (cancer cells, lymphocytes, macrophages, plasma cells, fibroblasts, and other cells or background). This collection of expert annotations was then used to train six-class cell-type and four-class tissue-type classifiers.

Several steps were taken to ensure the accuracy and generalizability of our models. First, it was important to recognize that common cell and tissue types, such as CAS or cancer cells, show morphological differences between BRCA, LUAD, LUSC, SKCM, and STAD. As a result, we trained separate cell- and tissue-type detection models for each of these five cancer types, for a total of ten models. Second, it was important to ensure that our models did not overfit to the histological patterns found in the training set. To avoid this, we followed the conventional protocol of splitting our data into training, validation, and test sets and incorporated additional annotations of the same five cancer types from PathAI's databases into the model development process. Together, these datasets represented a wide diversity of examples for each class in each cancer type, thus improving the generalizability of these models beyond the TCGA dataset.

Using the combined dataset of annotated TCGA and additional WSIs, we trained deep CNNs to output dense pixelwise cell- and tissue-type predictions at a subcellular spatial resolution of 2 and 4 μm, respectively (spatial resolution dictated by stride). To ensure that our models achieved sufficient accuracy for feature extraction, models were trained in an iterative process, with each updated model's predictions visualized as heatmaps to be reviewed by board-certified pathologists. In heatmap visualizations, tissue categories were segmented into colored regions, while cell types were identified as colored squares. This process continued until there were minimal systematic errors and the pathologists deemed the model sufficiently trustworthy for feature extraction.

All WSIs used in this study were FFPE slides. This means that tissue samples used for RNA-Seq and histology imaging were extracted from different portions of the patient's tumor and may thus vary in their TME.

**Pathologist-in-the-loop CNN model training**. During the CNN training process, we worked iteratively with three board-certified pathologists to conduct subjective evaluation of model predictions to inform multiple rounds of training. CNN models were initially trained on a set of primary annotations collected from the pathologist network. Following the conclusion of each training round (defined by model convergence), predicted cell and tissue heatmaps were reviewed for systematic errors (e.g., overprediction of fibroblasts, macrophages, and plasma cells, underprediction of necrotic tissue). New (secondary) annotations would then be collected from the pathologist network focusing on areas of improvement (e.g., mislabeled macrophages) to initiate a subsequent training round. The final cell- and tissue-type models were selected based on a consensus across the three pathologists. To reduce the risk of overfitting, CNN models were frozen after selection and unperturbed during molecular phenotype prediction using classical machine learning models. We computed validation metrics for cell- and tissue-type models on pooled primary and secondary annotations and visualized these metrics as confusion matrices.

**Pathologist validation of cell-type models**. To directly compare our cell-type predictions on TCGA WSIs against pathologist annotations, we generated 250 75 × 75 μm frames of cell-type overlays evenly sampled across the five cancer types and

five cell types, each from a distinct WSI. The generation process sought to sample frames with both high and low densities of a given cell-type according to our cell-type model predictions. Each frame was annotated for each of the five cell types by five board-certified pathologists. This allows us to compare the count of lymphocytes, plasma cells, fibroblasts, macrophages, and cancer cells in each $75 \times 75$ µm frame predicted by our CNN cell-type model against a consensus of pathologist annotation counts. We computed the Pearson correlation between our cell-type model counts and pathologist consensus counts across the 250 frames for all five cell types. Pathologist consensus counts were computed as the median of the five individual pathologist counts for a given frame and cell type. To capture inter-pathologist variability, we also computed the leave-one-out Pearson correlation between each individual pathologist's annotation counts and the consensus (median) among the remaining four pathologists. We then obtained a point estimate and 95% confidence interval for the average performance of an annotator with respect to the leave-one-out consensus.

To assess model generalizability, we redeployed our BRCA cell-type model trained primarily on TCGA to exhaustively predict cell types on 72 H&E, FFPE WSIs from an external BRCA dataset uploaded by Peikari et al. to TCIA[35]. We then used the same analysis framework and metrics as above to assess concordance between our cell-type model and pathologist consensus across 250 $75 \times 75$ µm frames (evenly sampled across the five cell types) generated from these external WSIs (Supplementary Fig. 4).

**Tissue-based feature extraction**. Using the tissue-type predictions, we extracted 163 different region-based features from each WSI in the TCGA dataset. Each of these features belonged to one of three general categories.

The first category consisted of areas ($n = 13$ HIFs). By simple pixel summation, we computed the total areas (in mm²) of cancer tissue, CAS, cancer tissue plus CAS, regions at the CSI, and necrosis in each slide. These features are interpretable and technically attainable by human pathologists but would be prohibitively time-consuming and inconsistent across pathologists to calculate in practice.

The second category, which contributed the bulk of the features, made use of the publicly available scikit-image.measure.regionprops module to find the connected components of each of these tissue types at the pixel-level using eight-connectivity. Once these connected components were found, we used both library-provided and self-implemented methods to extract a series of morphological features ($n = 125$ HIFs), similar to the approach suggested by Wang et al. in 2018[27]. These HIFs measured a wide variety of tissue characteristics, ranging from quantitative, size-based measures like the number of connected components, major and minor axis lengths, convex areas, and filled areas, to more qualitative, shape-based measures like Euler numbers, lacunarity, and eccentricity. Recognizing the log-distribution of connected component size, we computed these features not just across all connected components but also for both the largest connected component only and across the most "significant" connected components, defined as components >10% the size of the largest connected component. In aggregating metrics across considered components, we incorporated both averages and standard deviations of HIFs (e.g., standard deviation of eccentricities of significant regions of necrosis) to capture both summary metrics and metrics of intratumor heterogeneity.

The third category of features captures tissue architecture ($n = 25$ HIFs). Inspired by Lennon et al.[28], we calculated the fractal dimensions and solidity measures of different tissue types, capturing both the roundness and filled-ness of the tissue, under the hypothesis that the ability for these measures to separate different subtypes of lung cancer might translate to a similar ability to predict clinically relevant phenotypes. These features allowed us to capture information about how tissue filled up space, rather than just the summative sizes and shapes captured by the first and second categories.

**Cell- and tissue-based feature extraction**. After obtaining six-class cell-type predictions for each pixel of a WSI, we generated five binary masks corresponding to each of the five specified cell types. We then combined cell- and tissue-level masks to compute properties of each cell type in each tissue type (e.g., fibroblasts in CAS), extracting 444 HIFs.

An initial group of features that were readily calculable from our model predictions included simple counts and densities of cell types in different tissue types. For example, an overlay of a particular slide's lymphocyte detection mask on top of the same slide's CAS mask could be used to calculate the number of TILs on a given slide. We could then divide this number by the area of CAS to find the associated density of TILs on the slide. By taking the "outer product" of cell and tissue types, we derived a wide array of composite features. In particular, we calculated counts, proportions, and densities of cells across different tissue types (e.g., density of macrophages in CAS versus in cancer tissue), under the hypothesis that these measures capture information that raw counts could not. To capture information regarding cell–cell proximity and interactions, we also calculated counts and proportions of each cell type within an 80-µm radius of each other cell type (e.g., count of lymphocytes within an 80-µm radius of fibroblasts). Cell-level counts, densities, and proportions comprised 264 HIFs.

For each cell–tissue combination, we next applied the Birch clustering method (as implemented in the sklearn.cluster Python module) to partition cells into clusters[87]. To fit clustering structures as closely as possible to the spatial

relationships found between cell types on the slide, we set a threshold of 100, a branching factor of 10, and allowed the algorithm to optimize the number of clusters returned. We used the returned clusters to calculate a series of features designed to capture spatial relationships between individual cells types within a given tissue type, including number of clusters, cluster size mean and standard deviation (SD), within-cluster dispersion mean and SD, cluster extent mean and SD, the Ball–Hall Index, and Calinski–Harabasz Index ($n = 180$ HIFs). For metrics where cluster exemplars were needed, the subcluster centers returned by the Birch algorithm were used.

**Patient-level aggregation**. Patients with multiple tissue samples were represented by the single sample with the largest area of cancer tissue plus CAS, computed during tissue-based feature extraction. All subsequent analyses were conducted at the patient level.

**HIF clustering**. Due to underlying biological relationships as well as the HIF generation process, there is significant correlation structure between many of the features. This presents a challenge of feature selection as much of the information contained in one feature will also be present in another. It also makes it difficult to control for multiple hypothesis testing, because the underlying number of tested hypotheses is significantly fewer than the number of features computed.

To identify groups of correlated HIFs, we clustered features via hierarchical agglomerative clustering using complete linkage, a cluster cutoff of 0.95, and pairwise correlation distance ($1 -$ absolute Spearman correlation) as the distance metric. We defined a set of HIF clusters for each cancer type independently, as well as another set for pan-cancer analyses (Supplementary Data 1). Clustering correlated features allows us to summarize the true underlying number of tested hypotheses.

**Visualization of cancer types in HIF space**. UMAP was applied for dimensionality reduction and visualization of patient samples from the 607-dimension HIF space into two dimensions (using parameters: number of neighbors = 15, training epochs = 500, distance metric = Euclidean). The V-Measure was computed to compare BRCA, STAD, SKCM, and NSCLC (LUAD and LUSC combined) classes against clusters generated by $k$-means ($k = 4$) applied to the 2-D UMAP projection[36,37]. To quantify differences between cancer types, HIF values were normalized pan-cancer into $Z$-scores. Median $Z$-scores were then computed per cancer type across 20 HIFs, each representing 1 of the 20 HIF clusters defined pan-cancer. Representative HIFs were selected based on subjective interpretability and high variance across cancer types. To determine the statistical significance of median $Z$-scores that were greater in one cancer type relative to others, $P$ values were estimated with the one-sided Mann–Whitney $U$ test, considering NSCLC subtypes LUAD and LUSC as one type.

**Validation of HIFs against molecular signatures**. To validate the ability of HIFs to capture meaningful cell- and tissue-level information, we computed Spearman correlations between HIFs and four canonical immune signatures from the PanImmune dataset[45]: (1) leukocyte infiltration, (2) IgG expression, (3) TGF-β expression, and (4) wound healing. We also assessed HIF correlation to (5) angiogenesis signature, also derived from PanImmune, and (6) hypoxia score, derived from Buffa et al.[43,44]. All six molecular signatures were quantified by mapping mRNA sequencing reads against gene sets associated with the aforementioned known immune and gene expression signatures. To estimate the correlation between HIF clusters and immune signatures, we computed the median absolute Spearman correlation per cluster and combined $P$ values associated with individual correlations via the Empirical Brown's method[42]. To control the false discovery rate, combined $P$ values per cluster were then corrected using the Benjamini–Hochberg procedure[88]. Correlation analyses were conducted for cancer types collectively and individually, using HIF clusters defined across all cancer types for assessment of concordance.

In addition, image-based cell quantifications for leukocyte fraction, lymphocyte fraction, and plasma cell fraction were validated by Spearman correlation to their sequencing-based equivalents from matched TCGA tumor samples, computed using CIBERSORT (cell-type identification by estimating relative subsets of RNA transcripts)[45]. CIBERSORT uses an immune signature matrix for deconvolution of observed RNA-Seq read counts into estimates of relative contributions between 22 immune cell profiles[41].

**Molecular phenotype label curation**. To reduce bias and protect against overfitting, the molecular phenotypes assessed in this study were selected after the cell- and tissue-type models were frozen. PD-1, PD-L1, and CTLA-4 expression data for each cancer type were collected from the PanImmune dataset[45], while TIGIT expression data were collected from the National Cancer Institute Genomic Data Commons[78]. PD-1, PD-L1, CTLA-4, and TIGIT expression levels were quantified from mapped mRNA reads against genes PDCD1, CD274, CTLA-4, and TIGIT, respectively, and normalized as $Z$-scores across all cancer types in TCGA. HRD scores were collected from Knijnenburg et al.[89]. The HRD score was calculated as the sum of three components: (1) number of subchromosomal regions with allelic imbalance extending to the telomere, (2) number of chromosomal breaks between

adjacent regions of least 10 Mb (mega base pairs), and (3) number of loss of heterozygosity regions of intermediate size (at least 15 Mb but less than whole chromosome length). Continuous immune checkpoint protein expression and HRD scores were binarized to high versus low classes using Gaussian mixture model (GMM) clustering with unequal variance (Supplementary Fig. 9). The binary threshold was defined as the intersection of the empirical densities between the two GMM-defined clusters. To evaluate the extent to which prediction tasks were correlated, Pearson correlation and percentage agreement metrics were computed pan-cancer ($n = 1893$ patients) between the five molecular phenotypes in continuous and binarized form, respectively (Supplementary Fig. 12).

**Hold-out set definition by TCGA tissue source site**. TCGA provides tissue source site information, which denotes the medical institution or company that provided the patient sample. For each prediction task (described below), a hold-out set was defined as approximately 20–30% of patient samples obtained from sites not seen in the training set (Supplementary Table 4). This validation methodology enables us to demonstrate model generalizability across varying patient demographics and tissue collection processes intrinsic to different tissue source sites. Patient barcodes corresponding to hold-out and training sets are provided in Supplementary Data 2.

**Supervised prediction of molecular phenotypes**. We conducted supervised prediction of binarized high versus low expression of five clinically relevant phenotypes: (1) PD-1 expression, (2) PD-L1 expression, (3) CTLA-4 expression, (4) HRD score, and (5) TIGIT expression. Predictions were conducted pan-cancer as well as for cancer types individually. SKCM was excluded from prediction tasks 1 to 4 due to insufficient outcome labels (number of observations <100 for tasks 1–3; number of positive labels <10 for task 4). For each of the 26 prediction tasks, we trained a logistic sparse group lasso (SGL) model[90] tuned by nested cross-validation (CV) with three outer folds and five inner folds using the corresponding training set. SGL provides regularization at both an individual covariate (as in traditional lasso) and user-defined group level, thus encouraging group-wise and within-group sparsity. The HIF clusters defined per cancer type and pan-cancer (previously described) were inputted as groups. HIFs were normalized to mean = 0 and SD = 1. In accordance with nested CV, hyper-parameter tuning was conducted using the inner loops and mean generalization error and variance were estimated from the outer loops. The three tuned models, each trained on two of the three outer folds and evaluated on the third outer fold, were ensembled by averaging predicted probabilities for final evaluation (reported in Fig. 6a and Supplementary Table 5) on the hold-out set. Hold-out performance was evaluated by AUROC and AUPRC. To identify predictive features, beta values from the three outer fold models were averaged to obtain ensemble beta values per HIF (see Fig. 6b caption for more details).

**End-to-end model benchmarking**. To compare our HIF-based approach against conventional end-to-end models, we trained 26 distinct CNNs for each of the 26 molecular phenotype prediction tasks described above using single-instance learning. We used the computationally efficient ShuffleNet architecture and the same hyper-parameters described in Kather et al.[11] (batch size of 512, patch size of $512 \times 512$ pixels at 2 μm per pixel, 30 unfrozen layers, learning rate of $5 \times 10^{-5}$) without additional tuning. The same training and hold-out sets from HIF-based model development were used to ensure that AUROC metrics were comparable.

**Random forest model comparison**. Additionally, we compared the performance (AUROC and AUPRC) of HIF-based linear models against HIF-based random forest models. Hyperparameters were all set to defaults for all 26 molecular phenotype prediction tasks: number of trees = 500, number of variables randomly sampled as candidates at each split = 25 (square root of the number of features—607), minimum size of terminal nodes = 1. Random forest models account for interaction effects and can thus test the hypothesis that capturing interactions between the 607 HIFs can improve model performance[91]. Once again, we maintained the same training and hold-out sets used during HIF-based linear model development.

**Statistical analysis**. To compute 95% confidence intervals for each prediction task, we generated empirical distributions of AUROC and AUPRC metrics each consisting of 1000 bootstrapped metrics, as recommended by multiple sources[92]. Bootstrapped metrics were obtained by sampling with replacement from matched model predictions (probabilities) and true labels for the corresponding hold-out set and re-computing AUROC and AUPRC on these two bootstrapped vectors. P values for AUROC and AUPRC hold-out metrics were denoted as the probability either metric was <0.5 under the aforementioned empirical distributions and multiple-hypothesis-corrected across the 26 prediction tasks using the Benjamini–Hochberg procedure[88]. P values for ensemble beta values of predictive HIFs were computed using a permutation test with 1000 iterations. During each iteration, labels in the training set were permuted and the previously described training process of nested CV and ensembling was re-applied to generate a new set of ensemble beta values per HIF. P values for individual HIFs were then obtained by comparing beta values in the original ensemble model against the corresponding

null distribution of ensemble beta values. Individual HIF P values were combined into cluster-level P values via the Empirical Brown's method[42] and corrected using the Benjamini–Hochberg procedure[88]. Data analyses in this study used the programming languages Python version 3.7.4 and R version 3.6.2. Analysis code has been uploaded to public repositories[93].

**Reporting summary**. Further information on research design is available in the Nature Research Reporting Summary linked to this article.

## Data availability
Histopathology images from the Cancer Genome Atlas dataset are available at https://www.cancer.gov/about-nci/organization/ccg/research/structural-genomics/tcga. The Cancer Imaging Archive histopathology images used for external validation can be downloaded from https://doi.org/10.7937/TCIA.2019.4YIBTJNO. RNA-Seq quantifications for PD-1, PD-L1, and CTLA-4, estimates of relative contributions between 22 immune cell profiles from CIBERSORT, and quantifications for leukocyte infiltration, TGF-β, IgG, and wound healing signature were obtained from the PanImmune dataset: https://gdc.cancer.gov/about-data/publications/panimmune. RNA-Seq quantifications for TIGIT were obtained from the PanCanAtlas dataset: https://gdc.cancer.gov/about-data/publications/pancanatlas. HRD scores were obtained from the dataset shared by Knijnenburg et al.: https://gdc.cancer.gov/about-data/publications/PanCan-DDR-2018. All feature tables, as well as source code for reproducing correlational analyses and molecular predictions, are available at https://github.com/Path-AI/hif2gene/tree/master/data/hifs. Access to cell- and tissue-type heatmaps as well as usage of cell- and tissue-type classification models are available upon reasonable request to academic investigators without relevant conflicts of interest for non-commercial use who agree not to distribute the data. Access requests can be made to amaro.taylor@pathai.com.

## Code availability
Codes for cell- and tissue-type model training, inference, and feature extraction are not disclosed. Access requests for such code will not be considered to safeguard PathAI's intellectual property. However, access to cell- and tissue-type heatmaps as well as usage of cell- and tissue-type classification models are available upon reasonable request to academic investigators without relevant conflicts of interest for non-commercial use who agree not to distribute the data. The source code for all downstream data analyses and figure generation in this work are publicly available and can be downloaded from https://github.com/Path-AI/hif2gene (https://doi.org/10.5281/zenodo.4532238)[93]. Access requests and queries about code can be made to amaro.taylor@pathai.com.

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

## Acknowledgements

We are grateful to the software engineering and machine learning teams at PathAI, Inc. for developing the systems and pipelines used for model development and feature extraction. We also thank the Harvard-MIT Program in Health Sciences & Technology for enabling J.A.D., J.K.W., and W.F.C. to conduct this work toward satisfaction of their research theses. This work was funded by PathAI, Inc.

## Author contributions

J.A.D., J.K.W., W.F.C., A.T.-W., H.L.E., and A.H.B. conceived the project. J.A.D. and W.F.C. trained the models, generated heatmaps, and computed image features. A.L. and C.M. assisted with model troubleshooting and feature computation. S.K.R. and M.B.R. provided feedback on predicted heatmaps to enable iterative model improvements. B.G. and I.N.W. supervised collection of pathologist annotations for model training. S.C.G. and R.P. conducted concordance analyses between cell-type model predictions and pathologist annotations. J.K.W. and J.A.D. conducted statistical analysis of image feature associations. J.K.W. developed models for molecular phenotype prediction and image feature visualization. H.L.E. and A.T.-W. supervised model training and statistical analyses. All authors contributed to preparation of the manuscript.

## Competing interests

The authors declare the following competing interests: A.K. and A.H.B. are the cofounders of PathAI, Inc., a company that builds artificial intelligence tools for pathology. J.A.D., J.K.W., W.F.C., V.M., S.C.G., R.P., B.G., S.H., S.K.R., C.M., A.L., A.P., R.M., J.K.K., M.B.R., M.C.M., I.N.W., A.T.-W., and H.L.E. are currently, or were formerly, employed at PathAI, Inc.
