## [Peer Review File · Nature Communications]

Reviewers' Comments:

Reviewer #1:

Remarks to the Author:

All comments have been sufficiently addressed.

Reviewer #2:

Remarks to the Author:

The authors have addressed my previous concerns adequately.

The statement of novelty in their revision though is not accurate as many other papers have used or described a similar approach:

"Our study is the first to demonstrate the value of combining deep learning-based cell- and tissue-type classifications to compute image features that are both biologically-relevant and human-interpretable"

"Our novel HIF-based approach integrates the predictive power of deep learning with the interpretability of feature engineering, enabling incorporation of prior knowledge with comparable performance to end-to-end models."

Combining neural networks with engineered features is not a new or unproven idea in this field. This is discussed in reviews (PMIDs 31399699, 32411818) and in an original research papers (PMID 29051570, 31997849, 31997849). Yinyin Yuan's group has used this approach as well. Many groups recognize the relative strength of deep learning in tasks like segmentation and classification, and how engineered features can be layered on top of these to improve interpretability. Not all engineered features are human-interpretable but certainly those like group orientation of neoplastic nuclei are. The authors should search the literature and identify and cite similar papers if this is discussed and it should not be used as a point of novelty.

Reviewer #3:

Remarks to the Author:

In this manuscript, the authors sought to apply machine learning approaches to associate molecular patterns of cancers with pathology images obtained from The Cancer Genome Atlas (TCGA). The revision did not address many issues identified previously, and the presented approaches lack novelty or scientific significance, as previous reviewers pointed out. Below are my specific comments.

1. In the revised manuscript, the authors focused on using conventional machine learning methods to analyze pre-defined image features. As the authors discussed in the introduction section, the approach is not novel. Many recent publications have demonstrated the associations between image features and various molecular signatures in the cancer types presented in this paper. These results are summarized in review articles (such as *Curr Genet Med Rep.* 2019 Dec; 7(4):

208–213 and *Med Image Anal.* 2020 Sep 25;67:101813) published earlier. Pan-cancer analyses have been conducted and reported in various journals. Thus, the presented analyses may not provide much additional scientific or clinical insights.

2. The authors developed their models with the TCGA data, with only one model presented in the paper validated in another publicly available dataset from The Cancer Imaging Archive (TCIA). Since TCGA samples are processed centrally, which makes the partition based on tissue source sites a way of conducting held-out cross-validation, rather than external validation.

3. The amount of interactions their models can accommodate is quite limited, as these interactions need to be captured in the feature definition process and cannot be informed by the data. This limitation of conventional approaches motivated many quantitative researchers to move beyond simple models and develop models that enable high flexibility while provides ways to minimize overfitting. The recent development of advanced machine learning methods further took advantage of the compositional nature of images, which outperformed conventional methods in almost every image-based machine learning challenge. The gain from the proposed methods is not clear.

4. Recent studies identified many associations between histopathology and tumor microenvironments. For example, the highlighted association between CTLA-4 expression and the tumor microenvironment in the manuscript is previously reported in many studies (*Gastric Cancer.* 2016 Jan;19(1):42-52; *Cancer Immunol Immunother.* 2017 Nov;66(11):1449-1461; and *J Cancer.* 2020 Sep 9;11(21):6365-6375). Similar findings on the correlations between TIGIT expression and histopathology are reported in other studies. It is unclear what specifically did the presented analyses add.

5. Recent development of interpretable machine learning has improved beyond saliency maps. Model-agnostic methods, such as partial dependence plots, accumulated local effect plots, and permutation feature importance have revealed many insights from complicated machine learning models. The approach proposed by the authors works less well in many settings (such as the cell detection and region segmentation tasks using their datasets) as the authors noted, and may not be more interpretable than commonplace deep learning models.

6. The bootstrapping procedure described in the manuscript may not be sufficient for supporting statistical significance, since the presented bootstrapping method only re-sample the matched pairs of predictions and ground truth labels. As the number of bootstraps is arbitrary, increasing the number of bootstraps will artificially decrease the estimated 95% confidence interval (and perhaps achieve any significance level) without improving the model.

7. The inter-rater annotations agreement for macrophages, plasma cells, and fibroblasts are weak to moderate, which undermined their conclusions and the applicability of the developed models in future analyses. This issue may be particularly concerning since the cell types with lower agreements are the ones underpinning the highlighted associations.

8. The authors mentioned that their model training process was based on a proprietary software framework, and the details are not described throughout the manuscript. As a recent perspective article published in *Nature* (*Nature.* 2020 Oct;586(7829):E14-E16) pointed out, some machine learning approaches are unfortunately not reported in a transparent report or enable reproducibility, which greatly hindered the progress of the field. The details of the training process are perhaps the most important aspect of describing machine learning procedures, since that is how researchers get useful representations out of the raw data. When future researchers experience dataset shift due to changes in environmental factors of cancers or applying the model to under-represented populations not included in current studies, model adjustment or re-training will be needed. As such, only reporting the final validation results may not be sufficient for the actual usage of the models.

Reviewer #4:

Remarks to the Author:

In this study, the authors have used machine learning techniques to recapitulate a variety of immuno-oncology biomarkers from digital whole slide images (WSIs) of routinely stained H&E slides, representing 5 different tumor types. The novelty involves the creation of overlays of the WSIs with colored annotations reflecting the various cellular and architectural features recognized by their algorithms, resulting in "human-interpretable image figures (HIFs)", which a pathologist can review to assess the model's performance.

We thank the reviewer for this concise summary of our work.

The decision to use material from the TCGA as the basis to train the algorithms is both a strength and a weakness. Obviously, access to the detailed molecular data from the TCGA allows incorporation of these data into the ground truth for the study. On the other hand, the cases in the TCGA are overwhelmingly primary tumor samples, and although there is no detailed information about the primary versus metastasis source of the additional cases in this study, there are essentially no Stage IV cases included. It is possible that some aspects of the tumor microenvironment are specific to the local microenvironment (such as the example the authors cite of macrophages in the lung) and may not reflect the metastatic situation, which is of course the scenario when immunotherapy would most likely be considered. Also, the TCGA nature of the cases means that no further annotation beyond histologic review can be performed. Specifically, immunolabeling to more precisely define stromal and inflammatory cell types would be much better ground truth than pathologist annotation. As the authors acknowledge, distinction of certain cell types is very challenging using H&E stained images – especially those originally scanned at only 20X magnification – and this challenge raises questions about the value of the ground truth to distinguish fibroblasts from macrophages from other cell types not annotated (endothelial cells, smooth muscle cells, myofibroblasts, dendritic cells, etc.), which are present in the tumor-associated stroma.

We agree that the use of TCGA has both benefits and limitations. We have updated the manuscript to reflect these trade-offs. In particular, we note in our Discussion:

"Biopsy images submitted to the TCGA dataset suffer from selection bias towards more definitive diagnoses and primary tumors, that may not generalize well to ordinary clinical settings."

This statement would be stronger if the authors discussed the extent to which the likelihood that the methods would generalize to the setting of metastatic disease which is the current role for immune based therapies in most tumor types. Since this is the primary claim for future clinical utility of the methodologies in this manuscript, I believe this should be further clarified and will be important bit of clarification for a general audience as not to overstate the conclusions.

I presume that all of the analysis was performed on the TCGA FFPE material although this is not explicitly stated in the methods. For readers that are less familiar with the practice of pathology, it should be highlighted the extent to which the material analyzed is standard FFPE versus frozen section. My understanding of the methodology for the TCGA is that the frozen sections are adjacent the tissue that was sequenced whereas the FFPE sections are not adjacent so there is likely to be some variability between what is sequenced and the cellular components observed on the digital slide. Nothing can be done about this but it is worth mentioning.

We also expand upon the difficulty of disambiguating morphologically-similar cell types:

"morphologically similar cells (e.g. macrophages, dendritic cells, endothelial cells, pericytes, myeloid derived suppressor cells, and atypical lymphocytes) may all be captured under a single cell-type prediction... Collecting targeted annotations of morphologically-similar cell types may decrease noise in HIF estimates and improve performance."

This is the challenge of using human interpretable features. I have annotated many 1000s of cells at the single cell level for similar projects. I know that my personal accuracy for predicting certain cell types even with high quality 40X scans is going to be low. Certain cell types simply cannot be distinguished by morphology alone no matter what the resolution. I believe this statement accurately represents the fundamental limitation of this methodologies in the paper. This study represents a herculean effort for the participating pathologists, as the annotation was

conducted by manual slide review, with indication of architectural patterns and, impressively, individual stromal and immune cell types, which served as the ground truth for training the algorithms. How scalable is this sort of ground truth determination for computational pathology? We thank the reviewer for highlighting this important point. Collecting high-fidelity pathologist annotations for cell- and tissue-types can be time-consuming and expensive, which we agree may limit scalability. Obtaining such annotations is a challenge in the field of computational pathology. We have updated our Discussion to note:

“Second, curation of high-fidelity, large-scale pathologist annotations can be time-consuming and expensive. Improvement of available open-source segmentation models could accelerate the adoption of HIF-based models.”

I disagree with Reviewer 4 that this is a limitation as a well-equipped laboratory can achieve many annotations, perhaps not on the scale of this publication, but it is something we do in our laboratory with regularity. The preference for the community would be that the authors make at least some of the annotations open source but it is understandable for the authors to not given the cost incurred. Once open source annotations are available different models can be benchmarked as is commonly done for computer vision tasks. There are currently publicly available datasets for lymphocytes in H&E histopathology images for example. It would be valuable if the authors had benchmarked their performance in detecting lymphocytes with published machine learning models. We hope that our study, by demonstrating the value of cell-type and tissue-type, may help spur advancement of such models for further study.

The goal of computational algorithms such as these should be to better predict end points (outcome, response to therapy, etc.) than currently available technologies, or at least to predict the same outcomes more efficiently or reproducibly. By using pathologist annotations along with other imperfect biomarkers as the ground truth, it seems that it may not be possible to improve on the accuracy of these techniques to predict the outcome. If, instead, the clinical outcome (e.g., response to immunotherapy) were used as the ground truth, would there not be a greater chance the algorithms could identify some feature(s), embedded within the tumor morphology, that would better predict outcome than the morphology and existing biomarkers?

When I first became aware of this work and other similar projects, this is my primary criticism. Predicting a biomarker with an AUROC of X% that has an AUROC of Y% for the biologically relevant outcome, such as response to a specific immune checkpoint blockage agent, is truly a long way off from reaching clinical utility and it is possible that the relationship between what is being predicted by the HIF model has nothing to do with what might be biologically relevant for response to the therapy. Thus a disclaimer statement is warranted.

We agree that predicting biomarker and molecular phenotype outcomes is less clinically-relevant than predicting clinical endpoints and outcomes.

We chose to predict molecular phenotypes rather than clinical outcomes due to the lack of public datasets with matched high-resolution histopathology images and reliable clinical data. TCGA has high-quality images and molecule data, but its clinical data is far less reliable. We note in our updated Discussion:

“Third, TCGA has limited treatment data and clinical endpoint data is less reliable than molecular data.”

As the TCGA data is not several years old many of these cases predate the wide adoption of immune checkpoint blockage therapy. Thus for most of the patient’s there is not data regarding how they would have responded the therapies that are discussed.

With regards to clinical relevance, while we demonstrate that our HIF-based models can achieve robust performance in predicting clinically-actionable molecular phenotypes including PD-1, PD-L1, CTLA-4, HRD, and TIGIT that have clear linkages to clinical endpoints, we acknowledge that direct prediction of such endpoints may be necessary for our models to be used in clinical practice. As we note in our updated Discussion:

“While molecular phenotypes such as PD-L1 expression are informative for clinical endpoints such as sensitivity to immune checkpoint blockade^{7 8}, direct prediction of patient outcomes is needed for clinical integration. Our study provides an interpretable framework to generate hypotheses for clinically-relevant biomarkers that can be validated in future prospective studies.^{7 9} The curation of public datasets with matched pathology images and high-fidelity treatment information could

help bridge the remaining gap.”

This comment is appropriate.

We also agree that improved accuracy is required for clinical integration. We have revised the concluding sentence of our Discussion to read:

“With improved accuracy, HIF-based models could leverage this information to enable the discovery of novel patient subpopulations with specific treatment susceptibilities, novel biomarkers predictive of drug response, and hypotheses for subsequent research.”

There are emerging data using a variety of multiplexed immunomorphology platforms that provide much more precise spatial information about specific immune cell types involved in the modulation of response to immune-based therapy; the current study makes no attempt to replicate such data computationally (in part because these data cannot be derived from TCGA cases), but the potential for these technologies to predict immune response should be mentioned in the discussion for completeness.

We thank the reviewer for drawing the connection between our exhaustive cell- and tissue-type models and multiplex immunofluorescence technologies. In our Discussion we note:

“Future development of our approach could extend to multiplex immunofluorescence (mIF) technologies that measure spatial protein expression. These highly specific methods may improve upon traditional immunohistochemistry staining in predicting drug response to immune checkpoint inhibitors and eliminate the need for expert annotation of cell types.”

It is important to note that mIF methods are not scalable in the way that this manuscript’s methodology is. The typical region of interest analyzed by mIF is below WSI image scale and is too costly and noise-prone to analyze 1000s of slides.

Can the authors comment on spatial heterogeneity within individual tumors? For example, the immune response is typically most brisk at the interface of the tumor with the stroma.

We thank the reviewer for this suggestion. In our updated manuscript, we have included an analysis focused on assessing spatial heterogeneity among cell types at the cancer-stroma interface. In particular, we tested the density of each cell type within 80 microns of the cancer-stroma interface relative to within cancer tissue generally across the five cancer types. Our updated Results section states:

“Considering spatial heterogeneity, we observed an enrichment of lymphocytes and plasma cells in SKCM as well as an enrichment of cancer cells in LUSC and LUAD at the CSI relative to in cancer tissue plus cancer-associated stroma (CT+CAS) (Supplemental Figure 6).”

This is a nice addition to the manuscript. It might be possible that certain analyses restricted only to certain areas might add additional predictive power to the model but this seems to be out of scope for the current manuscript.

The abstract is lacking in data and other specifics.

We agree and have updated the abstract to include AUROC metrics, the number of molecular phenotypes predicted, the resolution of cell- and tissue-type models, and appended comparison between our HIF-based approach and end-to-end workflows.

Reviewer #5:

Remarks to the Author:

I have been asked by the editor to specifically comment on authors' responses to Reviewer 4 and whether the responses adequately address the reviewers concern. I must say that Review 4's critiques are almost verbatim with the comments that I had written as notes as I was reading the manuscript so I believe that other readers of similar background will raise many similar concerns regarding the methodology and utility of the work. That being said, this is a momentous amount of effort and as someone who is engaged in projects with similar objectives, I understand the extent to which you have gone to bring this manuscript forward.

My comments are the lines with ##

Reviewer #4:

Remarks to the Author:

In this study, the authors have used machine learning techniques to recapitulate a variety of immunology biomarkers from digital whole slide images (WSIs) of routinely stained H&E slides, representing 5 different tumor types. The novelty involves the creation of overlies of the WSIs with colored annotations reflecting the various cellular and architectural features recognized by their algorithms, resulting in "human-interpretable image figures (HIFs)", which a pathologist can review to assess the model's performance.

We thank the reviewer for this concise summary of our work.

The decision to use material from the TCGA as the basis to train the algorithms is both a strength and a weakness. Obviously, access to the detailed molecular data from the TCGA allows incorporation of these data into the ground truth for the study. On the other hand, the cases in the TCGA are overwhelmingly primary tumor samples, and although there is no detailed information about the primary versus metastasis source of the additional cases in this study, there are essentially no Stage IV cases included. It is possible that some aspects of the tumor microenvironment are specific to the local microenvironment (such as the example the authors cite of macrophages in the lung) and may not reflect the metastatic situation, which is of course the scenario when immunotherapy would most likely be considered. Also, the TCGA nature of the cases means that no further annotation beyond histologic review can be performed. Specifically, immunolabeling to more precisely define stromal and inflammatory cell types would be much better ground truth than pathologist annotation. As the authors acknowledge, distinction of certain cell types is very challenging using H&E stained images – especially those originally scanned at only 20X magnification – and this challenge raises questions about the value of the ground truth to distinguish fibroblasts from macrophages from other cell types not annotated (endothelial cells, smooth muscle cells, myofibroblasts, dendritic cells, etc.), which are present in the tumor-associated stroma.

We agree that the use of TCGA has both benefits and limitations. We have updated the manuscript to reflect these trade-offs. In particular, we note in our Discussion:

“Biopsy images submitted to the TCGA dataset suffer from selection bias towards more definitive diagnoses and primary tumors, that may not generalize well to ordinary clinical settings.”

This statement would be stronger if the authors discussed the extent to which the likelihood that the methods would generalize to the setting of metastatic disease which is the current role for immune based therapies in most tumor types. Since this is the primary claim for future clinical utility of the methodologies in this manuscript, I believe this should be further clarified and will be important bit of clarification for a general audience as not to overstate the conclusions.

I presume that all of the analysis was performed on the TCGA FFPE material although this is not explicitly stated in the methods. For readers that are less familiar with the practice of pathology, it should be highlighted the extent to which the material analyzed is standard FFPE versus frozen section. My understanding of the methodology for the TCGA is that the frozen sections are adjacent the tissue that was sequenced whereas the FFPE sections are not adjacent so there is likely to be some variability between what is sequenced an the cellular components observed on the digital slide. Nothing can be done about this but it is worth mentioning.

We also expand upon the difficulty of disambiguating morphologically-similar cell types:

“morphologically similar cells (e.g. macrophages, dendritic cells, endothelial cells, pericytes, myeloid derived suppressor cells, and atypical lymphocytes) may all be captured under a single cell-type prediction... Collecting targeted annotations of morphologically-similar cell types may decrease noise in HIF estimates and improve performance.”

This is the challenge of using human interpretable features. I have annotated many 1000s of cells at the single cell level for similar projects. I know that my personal accuracy for predicting certain cell types even with high quality 40X scans is going to be low. Certain cell types simply cannot be distinguished by morphology alone no matter what the resolution. I believe this statement accurately represents the fundamental limitation of this methodologies in the paper.

This study represents a herculean effort for the participating pathologists, as the annotation was conducted by manual slide review, with indication of architectural patterns and, impressively, individual stromal and immune cell types, which served as the ground truth for training the algorithms. How scalable is this sort of ground truth determination for computational pathology?

We thank the reviewer for highlighting this important point. Collecting high-fidelity pathologist annotations for cell- and tissue-types can be time-consuming and expensive, which we agree may limit scalability. Obtaining such annotations is a challenge in the field of computational pathology. We have updated our Discussion to note:

“Second, curation of high-fidelity, large-scale pathologist annotations can be time-consuming and expensive. Improvement of available open-source segmentation models could accelerate the adoption of HIF-based models.”

I disagree with Reviewer 4 that this is a limitation as a well-equipped laboratory can achieve many annotations, perhaps not on the scale of this publication, but it is something we do in our laboratory with regularity. The preference for the community would be that the authors make at least some of the annotations open source but it is understandable for the authors to not given the cost incurred. Once open source annotations are available different models can be benchmarked as is commonly done for computer vision tasks. There are currently publicly available datasets for lymphocytes in H&E histopathology images for example. It would be valuable if the authors had benchmarked their performance in detecting lymphocytes with published machine learning models.

We hope that our study, by demonstrating the value of cell-type and tissue-type, may help spur advancement of such models for further study.

The goal of computational algorithms such as these should be to better predict end points (outcome, response to therapy, etc.) than currently available technologies, or at least to predict the same outcomes more efficiently or reproducibly. By using pathologist annotations along with other imperfect biomarkers as the ground truth, it seems that it may not be possible to improve on the accuracy of these techniques to predict the outcome. If, instead, the clinical outcome (e.g., response to immunotherapy) were used as the ground truth, would there not be a greater chance the algorithms could identify some feature(s), embedded within the tumor morphology, that would better predict outcome than the morphology and existing biomarkers?

When I first became aware of this work and other similar projects, this is my primary criticism. Predicting a biomarker with an AUROC of X% that has an AUROC of Y% for the biologically relevant outcome, such as response to a specific immune checkpoint blockage agent, is truly a long way off from reaching clinical utility and it is possible that the relationship between what is being predicted by the HIF model has nothing to do with what might be biologically relevant for response to the therapy. Thus a disclaimer statement is warranted.

We agree that predicting biomarker and molecular phenotype outcomes is less clinically-relevant than predicting clinical endpoints and outcomes.

We chose to predict molecular phenotypes rather than clinical outcomes due to the lack of public datasets with matched high-resolution histopathology images and reliable clinical data. TCGA has high-quality images and molecule data, but its clinical data is far less reliable. We note in our updated Discussion:

“Third, TCGA has limited treatment data and clinical endpoint data is less reliable than molecular data.”

As the TCGA data is not several years old many of these cases predate the wide adoption of immune checkpoint blockage therapy. Thus for most of the patient’s there is not data regarding how they would have responded the therapies that are discussed.

With regards to clinical relevance, while we demonstrate that our HIF-based models can achieve robust performance in predicting clinically-actionable molecular phenotypes including PD-1, PD-L1, CTLA-4, HRD, and TIGIT that have clear linkages to clinical endpoints, we acknowledge that direct prediction of such endpoints may be necessary for our models to be used in clinical practice. As we note in our updated Discussion:

“While molecular phenotypes such as PD-L1 expression are informative for clinical endpoints such as sensitivity to immune checkpoint blockade^{7 8}, direct prediction of patient outcomes is needed for clinical integration. Our study provides an interpretable framework to generate hypotheses for clinically-relevant biomarkers that can be validated in future prospective studies.^{7 9} The curation of public datasets with matched pathology images and high-fidelity treatment information could help bridge the remaining gap.”

This comment is appropriate.

We also agree that improved accuracy is required for clinical integration. We have revised the concluding sentence of our Discussion to read:

“With improved accuracy, HIF-based models could leverage this information to enable the discovery of novel patient subpopulations with specific treatment susceptibilities, novel biomarkers predictive of drug response, and hypotheses for subsequent research.”

There are emerging data using a variety of multiplexed immunomorphology platforms that provide much more precise spacial information about specific immune cell types involved in the modulation of response to immune-based therapy; the current study makes no attempt to replicate such data computationally (in part because these data cannot be derived from TCGA cases), but the potential for these technologies to predict immune response should be mentioned in the discussion for completeness.

We thank the reviewer for drawing the connection between our exhaustive cell- and tissue-type models and multiplex immunofluorescence technologies. In our Discussion we note:

“Future development of our approach could extend to multiplex immunofluorescence (mIF) technologies that measure spatial protein expression. These highly specific methods may improve upon traditional immunohistochemistry staining in predicting drug response to immune checkpoint inhibitors and eliminate the need for expert annotation of cell types.”

It is important to note that mIF method are not scalable in the way that this manuscript’s methodology is. The typical region of interest analyzed by mIF is below WSI image scale and is too costly and noise-prone to analyze 1000s of slides.

Can the authors comment on spatial heterogeneity within individual tumors? For example, the immune response is typically most brisk at the interface of the tumor with the stroma.

We thank the reviewer for this suggestion. In our updated manuscript, we have included an analysis focused on assessing spatial heterogeneity among cell types at the cancer-stroma interface. In particular, we tested the density of each cell type within 80 microns of the cancer-stroma interface relative to within cancer tissue generally across the five cancer types. Our updated Results section states:

“Considering spatial heterogeneity, we observed an enrichment of lymphocytes and plasma cells in SKCM as well as an enrichment of cancer cells in LUSC and LUAD at the CSI relative to in cancer tissue plus cancer-associated stroma (CT+CAS) (Supplemental Figure 6).”

This is a nice addition to the manuscript. It might be possible that certain analyses restricted only to certain areas might add additional predictive power to the model but this seems to be out of scope for the current manuscript.

The abstract is lacking in data and other specifics.

We agree and have updated the abstract to include AUROC metrics, the number of molecular phenotypes predicted, the resolution of cell- and tissue-type models, and appended comparison between our HIF-based approach and end-to-end workflows.

Response to Reviewers

We thank the reviewers and editor for their thoughtful comments, which we have used to improve the manuscript. Below is a line-by-line response to comments from the four reviewers.

Reviewer #1 (Remarks to the Author):

All comments have been sufficiently addressed.

We thank the reviewer again for their helpful previous comments.

Reviewer #2 (Remarks to the Author):

The authors have addressed my previous concerns adequately.

The statement of novelty in their revision though is not accurate as many other papers have used or described a similar approach:

"Our study is the first to demonstrate the value of combining deep learning-based cell- and tissue-type classifications to compute image features that are both biologically-relevant and human-interpretable"

"Our novel HIF-based approach integrates the predictive power of deep learning with the interpretability of feature engineering, enabling incorporation of prior knowledge with comparable performance to end-to-end models."

Combining neural networks with engineered features is not a new or unproven idea in this field. This is discussed in reviews (PMIDs 31399699, 32411818) and in an original research papers (PMID 29051570, 31997849, 31997849). Yinyin Yuan's group has used this approach as well. Many groups recognize the relative strength of deep learning in tasks like segmentation and classification, and how engineered features can be layered on top of these to improve interpretability. Not all engineered features are human-interpretable but certainly those like group orientation of neoplastic nuclei are. The authors should search the literature and identify and cite similar papers if this is discussed and it should not be used as a point of novelty.

We agree that “combining neural networks with engineered features” is not a novel contribution; we intended to communicate novelty specifically for our implementation of [1] combined cell and tissue features computed across whole slide images (rather than patches), using [2] exhaustive high-resolution annotations, that [3] achieves comparable performance to fully “black-box” methods.

We have modified our Discussion, citing the publications above, to better contextualize our work in relation to prior studies:

“In recent years, fusion approaches that combine deep learning with feature engineering have gained traction.(Bera et al. 2019; Wang et al. 2017; Amgad et al. 2019; Amgad et al. 2020). Our study is among the first to combine exhaustive deep learning-based cell- and tissue-type classifications to compute image features that are both biologically-relevant and human-interpretable.”

“Notably, we show that our HIF-based approach, which integrates the predictive power of deep learning with the interpretability of feature engineering, achieves comparable performance to that of black-box models.”

Reviewer #3 (Remarks to the Author):

In this manuscript, the authors sought to apply machine learning approaches to associate molecular patterns of cancers with pathology images obtained from The Cancer Genome Atlas (TCGA). The revision did not address many issues identified previously, and the presented approaches lack novelty or scientific significance, as previous reviewers pointed out. Below are my specific comments.

We thank the reviewer for their comments and hope that our responses and modifications can more clearly describe the novelty and significance of our work.

1. In the revised manuscript, the authors focused on using conventional machine learning methods to analyze pre-defined image features. As the authors discussed in the introduction section, the approach is not novel. Many recent publications have demonstrated the associations between image features and various molecular signatures in the cancer types presented in this paper. These results are summarized in review articles (such as Curr Genet Med Rep. 2019 Dec; 7(4): 208–213 and Med Image Anal. 2020 Sep 25;67:101813) published earlier. Pan-cancer analyses have been conducted and reported in various journals. Thus, the presented analyses may not provide much additional scientific or clinical insights.

We thank the reviewer for sharing these two review articles. We have cited both in our updated Introduction. While prior approaches have demonstrated the ability to predict molecular signatures using either handcrafted or deep learning-derived image features, our HIF-based approach introduces several points of novelty.

Scale of interpretable features: as noted in our Introduction:

“While prior HIF-based studies have identified many useful feature classes, most have been limited in scope. Studies to date often involve a single cell or tissue type; none have explored features that combine both cell and tissue properties. In addition, the majority of reported HIFs have only been vetted on a single cancer type, often non-small-cell lung cancer (NSCLC).”

We build upon prior work by extending this HIF-based approach to a larger number of cancer types and generating a collection of 607 HIFs that systematically integrates cell- and tissue-type information.

Scale of annotations: we collected >1.6M cell- and tissue-type pathologist annotations across whole-slide images (WSIs). This level of detail allowed us to exhaustively generate cell- and tissue-type predictions at subcellular resolution (of two and four μm , respectively). In addition, we computed interpretable features that span entire whole slide images, thus improving on previously-described tiling methods that effectively downsample the image.

Comparable performance to end-to-end models: while we agree that “combining neural networks with engineered features” is not a novel contribution, the scale of annotations and interpretable features used in our HIF-based approach have allowed our models to achieve performance comparable to end-to-end approaches (Supplemental Table 6), demonstrating that expert-guided feature engineering can achieve state-of-the-art performance.

2. The authors developed their models with the TCGA data, with only one model presented in the paper validated in another publicly available dataset from The Cancer Imaging Archive (TCIA). Since TCGA samples are processed centrally, which makes the partition based on tissue source sites a way of conducting held-out cross-validation, rather than external validation.

We agree with the reviewer that our hold-out validation does not constitute fully external validation, and now comment on this limitation in our manuscript Discussion:

“Third, batch effects in TCGA can originate from differing tissue collection, sectioning, and processing procedures. Our validation procedure of partitioning by tissue source site does not account for all possible data artifacts, but it does control for sample collection, extraction, and other site-specific variables. Our HIF-based approach also limits the impact of spurious associations introduced by batch effects by pre-defining features based on biological phenomena.”

To directly assess model generalizability to external datasets, we redeployed our BRCA cell-type model trained primarily on TCGA to exhaustively predict cell types on 72 formalin-fixed,

paraffin-embedded (FFPE) WSIs stained using hematoxylin and eosin (H&E) from an external BRCA dataset uploaded by Peikari et al. to The Cancer Imaging Archive (TCIA). This analysis revealed robust concordance between our cell-type model and pathologist consensus in these external WSIs (Supplemental Figure 4).

3. The amount of interactions their models can accommodate is quite limited, as these interactions need to be captured in the feature definition process and cannot be informed by the data. This limitation of conventional approaches motivated many quantitative researchers to move beyond simple models and develop models that enable high flexibility while provides ways to minimize overfitting. The recent development of advanced machine learning methods further took advantage of the compositional nature of images, which outperformed conventional methods in almost every image-based machine learning challenge. The gain from the proposed methods is not clear.

We agree that interactions are often important for maximizing information capture. We report comparable performance between our HIF-based approach and end-to-end models, indicating that the relevant interactions are likely captured by our methods. To quantitatively evaluate the predictive value of additional interactions, we added a comparative analysis using random forest models, which efficiently capture complex feature interactions. Our Results section notes:

“While our HIF generation process explicitly encodes for interactions between biological entities... we also compared and achieved comparable hold-out AUROC and AUPRC performance between our HIF-based linear models against HIF-based random forest models, which directly account for interaction effects between HIFs (Supplemental Table 7).”

In many cases, the robustness against fitting to spurious interactions may also represent an advantage. Confounders such as tumor purity, histologic subtypes, or staining differences are not explicitly blocked from model fitting, which may lead to the end-to-end models presented using “secondary” or spurious signals with unclear clinical relevance. In contrast, our approach limits these signals by reducing the feature space.

We believe that HIF-based models hold several advantages over black box models. These gains include [1] improved model interpretability, [2] easy identification of failure modes, and [3] robustness to overfitting on spurious associations.

4. Recent studies identified many associations between histopathology and tumor microenvironments. For example, the highlighted association between CTLA-4 expression and the tumor microenvironment in the manuscript is previously reported in many studies (Gastric Cancer. 2016 Jan;19(1):42-52; Cancer Immunol Immunother. 2017 Nov;66(11):1449-1461; and J Cancer. 2020 Sep 9;11(21):6365-6375). Similar findings on the correlations between TIGIT

expression and histopathology are reported in other studies. It is unclear what specifically did the presented analyses add.

We agree that many reported relationships between HIFs and the tumor microenvironment have been previously documented in the literature. We sought to recapitulate prior findings that had been identified using well-accepted experimental methods as a means of validating our modeling approach. Additionally, while many previous associations have been suggested qualitatively, our HIF-based approach enables us to both validate and systematically quantify the strength of such associations. We have updated our Results to highlight this point:

“While many associations noted above have been previously identified using experimental methods, a HIF-based approach enables validation and systematic quantification of the strength of such associations.”

The primary value of the model, however, is discovery of many reported relationships that have not been previously documented or quantified. Besides the many associations between spatial and density features with PD-1, PD-L1, CTLA-4, TIGIT, and HRD scores reported in Figure 6, we added correlation analyses with two new gene expression signatures: angiogenesis and hypoxia (Supplemental Table 3 and Supplemental Figure 8). For both, we identify interesting HIF associations, including a novel relationship between “area of cancer-associated stroma relative to cancer tissue + cancer-associated stroma” and angiogenesis signature.

We thank the reviewer for sharing references regarding prior work on CTLA-4. Paulsen et al., Peng et al., and Kim et al. used a combination of bulk RNA-Seq and immunohistochemistry to identify associations between the tumor microenvironment and CTLA-4 expression. Our HIF-based approach improves on this work because it is able to capture more complex spatial features that require precise localization of cell types and tissue morphology. For instance, our HIF-based approach revealed that:

“The proximity of lymphocytes to cancer cells (pan-cancer and BRCA), morphology of necrotic regions (LUAD and LUSC), and density of cancer cells in CT+CAS versus exclusively in cancer-associated stroma (BRCA and STAD) were predictive of CTLA-4 expression across multiple models (Figure 6biii; Supplemental Figure 11iii).”

5. Recent development of interpretable machine learning has improved beyond saliency maps. Model-agnostic methods, such as partial dependence plots, accumulated local effect plots, and permutation feature importance have revealed many insights from complicated machine learning models.

We agree that interpretability in machine learning involves a large variety of methods that we have not specifically commented on, and have added this to our Discussion:

“Other model-agnostic interpretability methods, such as partial dependence plots and feature importance measures, are also unable to objectively and scalably connect pixel intensity features to biological phenomena. By contrast, predictive HIFs are directly mapped onto concepts of tissues and cells and can be interpreted quantitatively across thousands of images. This allows investigators to directly identify concrete hypotheses and correlations that can be investigated further in causal analyses.”

While saliency maps are the most common approach in computer vision for interpreting end-to-end classifiers, the model-agnostic methods mentioned above share the same fundamental disadvantage: they enable subjective generation of hypotheses but are not scalable and are susceptible to human biases.

Visualization methods like partial dependence plots and accumulated local effect plots are only feasible for qualitative comparisons and are not scalable for gigapixel images with billions of features. In addition, individual pixels are not comparable as the same “feature” between slides; this problem also applies to measures of feature importance. Image features also tend to be densely correlated, which is why methods like convolution have been so successful. These inter-feature correlations limit the use of partial dependence plots and most feature importance measures.

We believe that incorporating labels that explicitly map biological concepts (e.g., cells and tissues) is an important contribution of our work. This approach allows us to quantify and validate known relationships (e.g. density of fibroblasts in cancer-associated stroma with wound healing signature) while suggesting new ones (e.g. angiogenesis signature with the area proportion of cancer-associated stroma).

The approach proposed by the authors works less well in many settings (such as the cell detection and region segmentation tasks using their datasets) as the authors noted, and may not be more interpretable than commonplace deep learning models.

We agree that deep learning is unparalleled in prediction and segmentation tasks, such as cell and region classification. This is why our pipeline relies on convolutional neural networks for this step: generating information that are then combined into human-interpretable features. We believe simple linear models are more robust and interpretable for associating cell/tissue features with molecular phenotypes, but would never use linear models for the task of image classification from pixel data.

6. The bootstrapping procedure described in the manuscript may not be sufficient for supporting statistical significance, since the presented bootstrapping method only re-sample the matched pairs of predictions and ground truth labels. As the number of bootstraps is arbitrary, increasing the number of bootstraps will artificially decrease the estimated 95% confidence interval (and perhaps achieve any significance level) without improving the model.

We thank the reviewer for their comment. We believe there may be a misunderstanding regarding the bootstrapping procedure. Specifically, the 95% confidence interval reported is based on the distribution of observed AUROC metrics, rather than a statistical confidence for a central measure. Bootstrapped permutation testing generates confidence intervals that are asymptotically valid with higher sample counts. In other words, increasing the number of bootstraps should result in convergence toward the desired confidence interval, not progressively narrower intervals.

Below, we show empirical analyses from BRCA prediction of PD-1 expression validating our bootstrapped confidence intervals¹ across a range of sample counts (10^2-10^6) and relative to package-based implementations. We show our estimates [1] agree with standard implementations, [2] agree with non-bootstrap methods, and [3] are robust to the number of bootstrap iterations.

Number of Bootstrap Iterations	AUROC 95% Confidence Interval		
	Self-implemented bootstrap method ¹	pROC implementation ² : bootstrap method ¹	pROC implementation ² : DeLong method ³
100	0.716, 0.844	0.717, 0.842	0.718, 0.836
1,000	0.716, 0.835	0.716, 0.836	
10,000	0.718, 0.835	0.716, 0.834	
100,000	0.717, 0.834	0.716, 0.834	
1,000,000	0.717, 0.834	0.716, 0.834	

1. James Carpenter and John Bithell (2000) “Bootstrap confidence intervals: when, which, what? A practical guide for medical statisticians”. *Statistics in Medicine* 19, 1141–1164. DOI: 10.1002/(SICI)1097-0258(20000515)19:9<1141::AID-SIM479>3.0.CO;2-F.

2. Xavier Robin, Natacha Turck, Alexandre Hainard, et al. (2011) “pROC: an open-source package for R and S+ to analyze and compare ROC curves”. *BMC Bioinformatics*, 7, 77. DOI: 10.1186/1471-2105-12-77.
3. Elisabeth R. DeLong, David M. DeLong and Daniel L. Clarke-Pearson (1988) “Comparing the areas under two or more correlated receiver operating characteristic curves: a nonparametric approach”. *Biometrics* 44, 837–845.

7. The inter-rater annotations agreement for macrophages, plasma cells, and fibroblasts are weak to moderate, which undermined their conclusions and the applicability of the developed models in future analyses. This issue may be particularly concerning since the cell types with lower agreements are the ones underpinning the highlighted associations.

We thank the reviewer for this comment and agree that inter-pathologist agreement is far from perfect. Our analyses demonstrate that our model performance is comparable to inter-pathologist agreement across cell types (Supplemental Figure 3). Notably, inter-pathologist agreement represents the best performance that can be reasonably expected from models trained and evaluated using pathologist annotations as the ground truth. The accuracy of cell-type models is thus limited substantially by human disagreement. We have updated our Methods to clarify this point:

“As a benchmark, inter-pathologist correlation represents the optimal performance that can be expected from models trained and evaluated using pathologist annotations as the ground truth.”

Additionally, model performance depends only on the correlations between features and outcomes and is robust to linear scaling. For example, even if lymphocytes were overrepresented by a factor of two, the model would perform equally well so long as relative sample differences are preserved. In certain cases, morphologically-similar cell types may be classified together. We have addressed this limitation in our Discussion:

“HIFs may, in reality, capture information about a mixture of cell types. For example, in diffuse forms of STAD in which cancer cells invade smooth muscle tissue, our models misclassified certain smooth muscle cells as fibroblasts. Therefore, fibroblast-label HIFs likely reflect a mixture of these two cell types in STAD. In diffuse gastric cancer, immune cells and infiltrating cancer cells can also be difficult to disambiguate. Collecting targeted annotations of morphologically-similar cell types may decrease noise in HIF estimates and improve performance.”

8. The authors mentioned that their model training process was based on a proprietary software framework, and the details are not described throughout the manuscript. As a recent perspective

article published in Nature (Nature. 2020 Oct;586(7829):E14-E16) pointed out, some machine learning approaches are unfortunately not reported in a transparent report or enable reproducibility, which greatly hindered the progress of the field. The details of the training process are perhaps the most important aspect of describing machine learning procedures, since that is how researchers get useful representations out of the raw data. When future researchers experience dataset shift due to changes in environmental factors of cancers or applying the model to under-represented populations not included in current studies, model adjustment or re-training will be needed. As such, only reporting the final validation results may not be sufficient for the actual usage of the models.

We thank the reviewer for their comment. While we cannot release the deep learning models used for cell-type detection and tissue-type segmentation, we have published our full set of features (HIFs) for each cancer type at: <https://github.com/Path-AI/hif2gene/tree/master/data/hifs>. For validation of results, all analyses downstream of deep learning-based HIF-generation, including molecular phenotype prediction and correlation analyses, are included in the same repository.

We also agree that enabling access to model output can improve transparency. As such, we have also updated our data availability statement to note that access to cell and tissue heatmaps as well as usage of cell- and tissue-type models will be available upon reasonable request:

“All feature tables are available at: <https://github.com/Path-AI/hif2gene/tree/master/data/hifs>. Access to cell- and tissue-type heatmaps as well as usage of cell-type detection and tissue-type classification models are available upon reasonable request.”

Reviewer #5 (Remarks to the Author):

I have been asked by the editor to specifically comment on authors' responses to Reviewer 4 and whether the responses adequately address the reviewers concern. I must say that Review 4's critiques are almost verbatim with the comments that I had written as notes as I was reading the manuscript so I believe that other readers of similar background will raise many similar concerns regarding the methodology and utility of the work. That being said, this is a momentous amount of effort and as someone who is engaged in projects with similar objectives, I understand the extent to which you have gone to bring this manuscript forward.

My comments are the lines with ## (in dark red, bolded)

Thank you for filling in for Reviewer #4; we appreciate your additional input.

Reviewer #4:

Remarks to the Author:

In this study, the authors have used machine learning techniques to recapitulate a variety of immuno oncology biomarkers from digital whole slide images (WSIs) of routinely stained H&E slides, representing 5 different tumor types. The novelty involves the creation of overlies of the WSIs with colored annotations reflecting the various cellular and architectural features recognized by their algorithms, resulting in “human-interpretable image figures (HIFs)”, which a pathologist can review to assess the model’s performance.

We thank the reviewer for this concise summary of our work.

The decision to use material from the TCGA as the basis to train the algorithms is both a strength and a weakness. Obviously, access to the detailed molecular data from the TCGA allows incorporation of these data into the ground truth for the study. On the other hand, the cases in the TCGA are overwhelmingly primary tumor samples, and although there is no detailed information about the primary versus metastasis source of the additional cases in this study, there are essentially no Stage IV cases included. It is possible that some aspects of the tumor microenvironment are specific to the local microenvironment (such as the example the authors cite of macrophages in the lung) and may not reflect the metastatic situation, which is of course the scenario when immunotherapy would most likely be considered. Also, the TCGA nature of the cases means that no further annotation beyond histologic review can be performed. Specifically, immunolabeling to more precisely define stromal and inflammatory cell types would be much better ground truth than pathologist annotation. As the authors acknowledge, distinction of certain cell types is very challenging using H&E stained images – especially those originally scanned at only 20X magnification – and this challenge raises questions about the value of the ground truth to distinguish fibroblasts from macrophages from other cell types not annotated (endothelial cells, smooth muscle cells, myofibroblasts, dendritic cells, etc.), which are present in the tumor-associated stroma.

We agree that the use of TCGA has both benefits and limitations. We have updated the manuscript to reflect these trade-offs. In particular, we note in our Discussion: “Biopsy images submitted to the TCGA dataset suffer from selection bias towards more definitive diagnoses and primary tumors, that may not generalize well to ordinary clinical settings.”

This statement would be stronger if the authors discussed the extent to which the likelihood that the methods would generalize to the setting of metastatic disease which is the current role for immune based therapies in most tumor types. Since this is the primary claim for future clinical utility of the methodologies in this manuscript, I believe this should be further clarified and will be an important bit of clarification for a general audience as not to overstate the conclusions.

We agree with the reviewer that the generalizability of results to metastatic settings should be clearly addressed, and have highlighted this important caveat in our updated Discussion:

“Our study data from TCGA carries several limitations. First, biopsy images submitted to the TCGA dataset are biased towards primary tumors and tumors with more definitive diagnoses that may not generalize well to ordinary clinical settings. Indeed, associations identified in primary tumors may not necessarily translate to metastatic settings (Supplemental Figure 5).”

I presume that all of the analysis was performed on the TCGA FFPE material although this is not explicitly stated in the methods. For readers that are less familiar with the practice of pathology, it should be highlighted the extent to which the material analyzed is standard FFPE versus frozen section. My understanding of the methodology for the TCGA is that the frozen sections are adjacent tissue that was sequenced whereas the FFPE sections are not adjacent so there is likely to be some variability between what is sequenced and the cellular components observed on the digital slide. Nothing can be done about this but it is worth mentioning.

We thank the reviewer for highlighting the distinctions between FFPE and frozen samples. We have updated the Methods to more clearly note that FFPE (not frozen) samples were used:

“All WSIs used in this study were FFPE slides. This means that tissue samples used for RNA sequencing and histology imaging were extracted from different portions of the patient’s tumor, and may thus vary in their TME.”

We also expand upon the difficulty of disambiguating morphologically-similar cell types: “morphologically similar cells (e.g. macrophages, dendritic cells, endothelial cells, pericytes, myeloid-derived suppressor cells, and atypical lymphocytes) may all be captured under a single cell-type prediction... Collecting targeted annotations of morphologically-similar cell types may decrease noise in HIF estimates and improve performance.”

This is the challenge of using human interpretable features. I have annotated many 1000s of cells at the single cell level for similar projects. I know that my personal accuracy for predicting certain cell types even with high quality 40X scans is going to be low. Certain cell types simply cannot be distinguished by morphology alone no matter what the resolution. I believe this statement accurately represents the fundamental limitation of this methodologies in the paper.

We are glad to hear that our statement accurately portrays this limitation.

This study represents a herculean effort for the participating pathologists, as the annotation was

conducted by manual slide review, with indication of architectural patterns and, impressively, individual stromal and immune cell types, which served as the ground truth for training the algorithms. How scalable is this sort of ground truth determination for computational pathology?

We thank the reviewer for highlighting this important point. Collecting high-fidelity pathologist annotations for cell- and tissue-types can be time-consuming and expensive, which we agree may limit scalability. Obtaining such annotations is a challenge in the field of computational pathology. We have updated our Discussion to note: “Second, curation of high-fidelity, large-scale pathologist annotations can be time-consuming and expensive. Improvement of available open-source segmentation models could accelerate the adoption of HIF-based models.”

I disagree with Reviewer 4 that this is a limitation as a well-equipped laboratory can achieve many annotations, perhaps not on the scale of this publication, but it is something we do in our laboratory with regularity. The preference for the community would be that the authors make at least some of the annotations open source but it is understandable for the authors to not given the cost incurred. Once open source annotations are available different models can be benchmarked as is commonly done for computer vision tasks. There are currently publicly available datasets for lymphocytes in H&E histopathology images for example. It would be valuable if the authors had benchmarked their performance in detecting lymphocytes with published machine learning models.

We thank the reviewer for this comment.

Regarding open-source: we agree that implementing the same pipeline with tissue and cell models trained on extensive annotation is costly but feasible for many laboratories. While we cannot make our annotations open-source, we released our full set of features (HIFs) for each cancer type at: <https://github.com/Path-AI/hif2gene/tree/master/data/hifs>. We hope this can help laboratories validate our reported associations and extend our work using TCGA.

We have also updated our data availability statement to note that access to cell and tissue heatmaps as well as usage of cell- and tissue-type models will be available upon reasonable request:

“Access to cell- and tissue-type heatmaps as well as usage of cell-type detection and tissue-type classification models are available upon reasonable request.”

Regarding the validity of cell labeling: we agree that benchmarking on public lymphocyte H&E data would be helpful. However, because our models are trained to detect across five cell types,

we wanted to use a uniform method to evaluate validity for all of these, instead of one only available for lymphocytes. To do this, we benchmarked performance of our cell-type models against pathologist consensus on 72 FFPE/H&E WSIs from The Cancer Imaging Archive (TCIA), an external BRCA dataset uploaded by Peikari et al. Our analysis (Supplemental Figure 4) demonstrated robust concordance between our BRCA cell-type model and pathologist consensus; correlation coefficients ranged from 0.607 in macrophages to 0.926 in lymphocytes.

We hope that our study, by demonstrating the value of cell-type and tissue-type, may help spur advancement of such models for further study.

The goal of computational algorithms such as these should be to better predict end points (outcome, response to therapy, etc.) than currently available technologies, or at least to predict the same outcomes more efficiently or reproducibly. By using pathologist annotations along with other imperfect biomarkers as the ground truth, it seems that it may not be possible to improve on the accuracy of these techniques to predict the outcome. If, instead, the clinical outcome (e.g., response to immunotherapy) were used as the ground truth, would there not be a greater chance the algorithms could identify some feature(s), embedded within the tumor morphology, that would better predict outcome than the morphology and existing biomarkers?

When I first became aware of this work and other similar projects, this is my primary criticism. Predicting a biomarker with an AUROC of X% that has an AUROC of Y% for the biologically relevant outcome, such as response to a specific immune checkpoint blockage agent, is truly a long way off from reaching clinical utility and it is possible that the relationship between what is being predicted by the HIF model has nothing to do with what might be biologically relevant for response to the therapy. Thus a disclaimer statement is warranted.

We agree with the reviewer that while we demonstrate robust performance in predicting clinically-actionable molecular phenotypes (e.g. PD-1, PD-L1, CTLA-4, HRD, and TIGIT expression), this does not necessarily translate into the ability to robustly predict clinical endpoints, such as response to therapy. As such, we have modified our Discussion to note:

“While molecular phenotypes such as PD-L1 expression are informative for clinical endpoints such as sensitivity to immune checkpoint blockade, the ability to robustly predict biomarkers does not necessarily translate into robust prediction of relevant endpoints. Ultimately, direct prediction of patient outcomes is needed for clinical integration.”

We agree that predicting biomarker and molecular phenotype outcomes is less clinically-relevant than predicting clinical endpoints and outcomes. We chose to predict molecular phenotypes rather than clinical outcomes due to the lack of public datasets with matched

high-resolution histopathology images and reliable clinical data. TCGA has high-quality images and molecular data, but its clinical data is far less reliable. We note in our updated Discussion: “Third, TCGA has limited treatment data and clinical endpoint data is less reliable than molecular data.”

As the TCGA data is not several years old many of these cases predate the wide adoption of immune checkpoint blockage therapy. Thus for most of the patient’s there is not data regarding how they would have responded the therapies that are discussed.

We agree that this is an important detail to note. We have updated our Discussion to note:

“As TCGA samples were made available in 2013, treatment regimens for these cases also predate the widespread adoption of immune checkpoint inhibitors”

With regards to clinical relevance, while we demonstrate that our HIF-based models can achieve robust performance in predicting clinically-actionable molecular phenotypes including PD-1, PD-L1, CTLA-4, HRD, and TIGIT that have clear linkages to clinical endpoints, we acknowledge that direct prediction of such endpoints may be necessary for our models to be used in clinical practice. As we note in our updated Discussion: “While molecular phenotypes such as PD-L1 expression are informative for clinical endpoints such as sensitivity to immune checkpoint blockade, direct prediction of patient outcomes is needed for clinical integration. Our study provides an interpretable framework to generate hypotheses for clinically-relevant biomarkers that can be validated in future prospective studies.^{7,9} The curation of public datasets with matched pathology images and high-fidelity treatment information could help bridge the remaining gap.”

This comment is appropriate.

We also agree that improved accuracy is required for clinical integration. We have revised the concluding sentence of our Discussion to read: “With improved accuracy, HIF-based models could leverage this information to enable the discovery of novel patient subpopulations with specific treatment susceptibilities, novel biomarkers predictive of drug response, and hypotheses for subsequent research.”

There are emerging data using a variety of multiplexed immunomorphology platforms that provide much more precise spatial information about specific immune cell types involved in the modulation of response to immune-based therapy; the current study makes no attempt to replicate such data computationally (in part because these data cannot be derived from TCGA cases), but the potential for these technologies to predict immune response should be mentioned in the discussion for completeness.

We thank the reviewer for drawing the connection between our exhaustive cell- and tissue-type models and multiplex immunofluorescence technologies. In our Discussion we note: “Future development of our approach could extend to multiplex immunofluorescence (mIF) technologies that measure spatial protein expression. These highly specific methods may improve upon traditional immunohistochemistry staining in predicting drug response to immune checkpoint inhibitors and eliminate the need for expert annotation of cell types.”

It is important to note that mIF method are not scalable in the way that this manuscript’s methodology is. The typical region of interest analyzed by mIF is below WSI image scale and is too costly and noise-prone to analyze 1000s of slides.

We agree with the reviewer on this point and have modified our statement in the Discussion to reflect these considerations:

“Future development of our approach could extend to multiplex immunofluorescence (mIF) technologies that measure spatial protein expression. These methods face challenges of increased cost, lower resolution, and lower scalability across WSIs, but may improve upon traditional immunohistochemistry staining in predicting drug response to immune checkpoint inhibitors and reduce the need for expert annotation of cell types.”

Can the authors comment on spatial heterogeneity within individual tumors? For example, the immune response is typically most brisk at the interface of the tumor with the stroma.

We thank the reviewer for this suggestion. In our updated manuscript, we have included an analysis focused on assessing spatial heterogeneity among cell types at the cancer-stroma interface. In particular, we tested the density of each cell type within 80 microns of the cancer-stroma interface relative to within cancer tissue generally across the five cancer types. Our updated Results section states: “Considering spatial heterogeneity, we observed an enrichment of lymphocytes and plasma cells in SKCM as well as an enrichment of cancer cells in LUSC and LUAD at the CSI relative to in cancer tissue plus cancer-associated stroma (CT+CAS) (Supplemental Figure 6).”

This is a nice addition to the manuscript. It might be possible that certain analyses restricted only to certain areas might add additional predictive power to the model but this seems to be out of scope for the current manuscript.

Our analysis currently includes “localized” computation of HIFs within each defined tissue region (e.g., cancer tissue, cancer-associated stroma), but we agree that finer localizations

methods (e.g., patch-based HIF-distributions within tissue types) would be a great area to pursue further. We have updated the Discussion to note such “attention-based” methods as a future direction:

“Lastly, HIFs are computed as summary statistics within each tissue type across WSIs. Applying “attention-based” HIF computation to focus on regions of interest and further account for spatial heterogeneity is a potential avenue for further research.”

The abstract is lacking in data and other specifics.

We agree and have updated the abstract to include AUROC metrics, the number of molecular phenotypes predicted, the resolution of cell- and tissue-type models, and appended comparison between our HIF based approach and end-to-end workflows.

Reviewers' Comments:

Reviewer #3:

Remarks to the Author:

Thank you for your response. While the revision revealed some approaches employed by the authors, a few key issues regarding the reliability of the conclusions and the contribution of the work given the non-transparent reporting persists. Below are my comments.

1. Since the features defined by the proposed approach relied on the accurate identification of cell types, the low-to-moderate inter-rater agreement in identifying immune cells and fibroblasts may substantially impact the reliability of the conclusions, as the authors noted. What are the causes of the modest inter-rater agreement in these cell types? When pathologists disagree on the annotations, how did the trained models behave?

2. The p-value and q-value derived from the bootstrap may still suffer from the arbitrary number of bootstrap iterations. Other things being equal, p-values are known to reduce when the sample size increases. Based on the current descriptions of their bootstrap methods (an arbitrary number of bootstraps, according to the table, followed by a conventional statistical test with multiple test correction, as presented in the methods section), it can inflate the statistical power to any arbitrary level. For example, when researchers follow the same approach and decide to run one billion iterations, any non-random model may achieve statistical significance.

3. Using the 607 features proposed here may not avoid identifying spurious associations. As the authors pointed out, there are many potential confounders in the presented analyses, such as staining difference, tumor purity, and histologic subtypes. However, none of these confounders were controlled or adjusted in any of the presented models, according to the reported methods. Thus, if potential spurious associations are of concern, they will likely persist in these results. Simple models (such as generalized linear models) do not preclude or mitigate the effects of confounders. In addition, the effects of these confounders were not assessed in the study.

4. The deep learning model that serves as a baseline of comparison was not designed specifically for the task and is not tuned. As modern machine learning architectures require substantial fine-tuning, it is possible that the current baseline model is not representative of the machine learning models researchers will use for similar tasks.

5. As pointed out by previous reviewers, the authors attempted to validate their models, but only one analysis is validated with another publicly-available dataset. The robustness of the remaining models is unknown.

6. Recent pan-cancer analyses on TCGA histopathology data involved more than 20 types of cancers available from the Genomic Data Commons platform and reported the associations between oncogenes, tumor suppressor genes, and digital pathology features. Thus the analyses of the selected cancer types are not new and are covered by these prior studies.

7. The authors stated that the scale of the number of interpretable features ($m=607$) as a novelty of the study. However, many previous studies cited in the introduction section used several thousand interpretable pre-defined features via a computational framework. It is unclear if the scale is truly a highlight.

8. The current reporting of methods is still unfortunately opaque. It is impossible for readers to reproduce the proposed features from the histopathology images or to improve the feature set. The hyperparameters and model training procedures for the cell detection model are not available either. The lack of such information makes it difficult to satisfy reporting guidelines regarding machine learning applications.

Reviewer #5:

Remarks to the Author:

I am satisfied by the responses to my specific comments and do not have any further concerns regarding the publication of the manuscript.

Response to Reviewers

We thank the reviewers and editor for their thoughtful comments, which we have used to improve the manuscript. Below is our response to new comments from reviewers 1, 3, and 5.

Reviewer #1

I read Reviewer #3's feedback with great interest. He or she is very attentive to details and rightly pointed out a few minor issues. For example, points 2, 5 and 6 certainly relevant and should be addressed by a comment in the manuscript. Also, I agree with point 8, stating that the transparency of the methods reporting should be still further improved. However, I do not think that the points 1, 3, 4 and 7 are really helpful - the reviewer is being very strict here which I am not sure is fair. Overall I think that this study is of high scientific and technological quality and is should be eventually published once the above-mentioned minor points are addressed."

Editors

Please refer to these comments to prepare your last responses to Reviewer 3 as we agree with Reviewer 1.

We thank the reviewer and editors for identifying the minor points remaining. We have focused particular attention on addressing points 2, 5, 6, and 8 below.

Additional software/ algorithms, for e.g. Birch and CIBERSORT were observed in the manuscript. Please list all the software used for data collection/analysis in the study, along with their version numbers, in the reporting summary as well.

Birch clustering was one of dozens of statistical techniques implemented in standard Python libraries (in this case, scikit-learn), not a distinct software. The reporting summary now notes all libraries used for Python and R.

CIBERSORT was used to generate quantifications in the PanImmune dataset (Thorsson et al. 2018) that we used in our analyses, but we did not run CIBERSORT ourselves. As a result, we have listed PanImmune in our data availability statement, but not among software and code.

Please ensure that information on datasets used from published sources is provided under the 'data' section of the reporting summary, as the information is not relevant to this field. Also, please list the software used for data collection OR if no software was used for data collection, please state here accordingly.

We have ensured that information from published sources are covered by the data availability statement and in the data section of the reporting summary. No software was used for data collection; we now state this in the reporting summary.

Please provide the version numbers of the following software in the reporting summary: Python and R. Please list these software python and R in the manuscript, since they have been referenced here in the reporting summary.

We have included the following statement in the Methods section of the manuscript and in the reporting summary:

“Data analyses in this study used the programming languages Python version 3.7.4 and R version 3.6.2.”

Please provide a complete data availability statement, here in the reporting summary, as well as in the manuscript by combining the statements currently provided here in the reporting summary and in the manuscript.

We have included the following data availability statement in the manuscript and in the reporting summary:

“Histopathology images from the Cancer Genome Atlas dataset are available at <https://www.cancer.gov/about-nci/organization/ccg/research/structural-genomics/tcga>. The Cancer Imaging Archive histopathology images used for external validation can be downloaded from: <https://doi.org/10.7937/TCIA.2019.4YIBTJNO>. RNASeq quantifications for PD-1, PD-L1, and CTLA-4, estimates of relative contributions between 22 immune cell profiles from CIBERSORT, and quantifications for leukocyte infiltration, TGF-Beta, IgG, and wound healing signature were obtained from the PanImmune dataset: <https://gdc.cancer.gov/about-data/publications/panimmune>. RNASeq quantifications for TIGIT were obtained from the PanCanAtlas dataset: <https://gdc.cancer.gov/about-data/publications/pancanatlas>. HRD scores were obtained from the dataset shared by Knijnenburg et al: <https://gdc.cancer.gov/about-data/publications/PanCan-DDR-2018>. All feature tables, as well as source code for reproducing correlational analyses and molecular predictions, are available at: <https://github.com/Path-AI/hif2gene/tree/master/data/hifs>. Access to cell- and tissue-type heatmaps as well as usage of cell-type and tissue-type classification models are available upon reasonable request to academic investigators without relevant conflicts of interest who agree not to distribute the data.”

Please state how often the experiments were replicated or performed independently.

We have updated the “replication” entry as follows:

“Evaluation metrics were replicated across resampled data in order to derive empirical distributions for these metrics. This replication then served as the basis for confidence intervals, p-values, and q-values.”

Please provide a rationale for why blinding was not performed in this study as blinding procedures are also applicable for experiments other than animals and human research subjects.

We have updated the “blinding” entry as follows:

“Research personnel working on training tissue- and cell-type models were blinded to the molecular phenotype data and evaluation metrics. Models evaluated on held-out cohorts were blinded to the data and distributions of those cohorts.”

Reviewer #3

Thank you for your response. While the revision revealed some approaches employed by the authors, a few key issues regarding the reliability of the conclusions and the contribution of the work given the non-transparent reporting persists. Below are my comments.

1. Since the features defined by the proposed approach relied on the accurate identification of cell types, the low-to-moderate inter-rater agreement in identifying immune cells and fibroblasts may substantially impact the reliability of the conclusions, as the authors noted. What are the causes of the modest inter-rater agreement in these cell types? When pathologists disagree on the annotations, how did the trained models behave?

We agree with reviewer #1 and the editors that this point has already been sufficiently addressed, but hope that the following response can provide further clarity.

First, our evaluations suggest that cell-type model predictions reliably correlate with pathologist consensus:

“We observed that Pearson correlations between cell-type model predictions and pathologist consensus were comparable to inter-pathologist correlation (differences in correlation ranged from -0.019 to 0.024, with a median absolute difference of 0.069) across the five cell types... As a benchmark, inter-pathologist correlation represents the optimal performance that can be expected from models trained and evaluated using pathologist annotations as the ground truth.” (Methods)

Causes of inter-rater disagreement include morphologic similarity of different cell types and limits of H&E; both are addressed in the limitations section of the Discussion:

- “morphologically similar cells (e.g. macrophages, dendritic cells, endothelial cells, pericytes, myeloid derived suppressor cells, and atypical lymphocytes) may all be captured under a single cell-type prediction. Thus, HIFs may, in reality, capture information about a mixture of cell types. For example, in diffuse forms of STAD in which cancer cells invade smooth muscle tissue, our models misclassified certain smooth muscle cells as fibroblasts. Collecting targeted annotations of morphologically-similar cell types may decrease noise in HIF estimates and improve performance.”
- “Macrophages are particularly difficult for pathologists to identify solely under H&E staining. While the accuracy of an individual pathologist identifying macrophages may be poor, our models represent an aggregate estimate based on training from hundreds of pathologist annotators, which may carry a more reliable signal^{83,84}.”

Regarding pathologist disagreement: models are trained on a large number of annotations from hundreds of different pathologists, and so are expected to predict the consensus when individual pathologists disagree (as observed in our correlation analyses).

2. The **p-value and q-value derived from the bootstrap** may still suffer from the arbitrary number of bootstrap iterations. Other things being equal, p-values are known to reduce when the sample size increases. Based on the current descriptions of their bootstrap methods (an arbitrary number of bootstraps, according to the table, followed by a conventional statistical test with multiple test correction, as presented in the methods section), it can inflate the statistical power to any arbitrary level. For example, when researchers follow the same approach and decide to run one billion iterations, any non-random model may achieve statistical significance.

To meet the interest of the editors and the reviewer, we have recomputed all AUROC confidence intervals, p-values, and q-values in Supplementary Tables 5-7 with non-bootstrap values derived from the DeLong method, implemented by the R package pROC. There was no significant difference between the outputs from bootstrap and the DeLong method.

To address the reviewer’s comments: we did not pick an arbitrary number of bootstraps. We followed the guidance of Carpenter and Bithell. *Statist. Med.* 2000¹, which is also the source that our package implementation (pROC) replies on.

“2.4 How many bootstrap samples

A key question faced by anyone using the bootstrap is how large should B be. For 90-95 percent confidence intervals, most practitioners (for example, Efron and Tibshirani, Reference [6], p. 162, Davison and Hinkley, Reference [5], p. 194) suggest that B should be between 1000 and 2000.”

We have updated our methods to reflect this detail:

“To compute 95% confidence intervals for each prediction task, we generated empirical distributions of AUROC and AUPRC metrics each consisting of 1000 bootstrapped metrics, as recommended by multiple sources.⁹³”

We hope this explanation, along with our updated tables, can support reviewer and editorial confidence in our data. In addition, we have copied prior analyses showing broad agreement of our estimates across methods (DeLong vs. bootstrap), sample counts (100-1M), and public software implementations (pROC).

Number of Bootstrap Iterations	BRCA prediction of PD-1: AUROC 95% Confidence Interval		
	Self-implemented bootstrap method ¹	pROC implementation ² : bootstrap method ¹	pROC implementation ² : DeLong method ³
100	0.716, 0.844	0.717, 0.842	0.718, 0.836
1,000	0.716, 0.835	0.716, 0.836	
10,000	0.718, 0.835	0.716, 0.834	
100,000	0.717, 0.834	0.716, 0.834	
1,000,000	0.717, 0.834	0.716, 0.834	

1. James Carpenter and John Bithell (2000) “Bootstrap confidence intervals: when, which, what? A practical guide for medical statisticians”. *Statistics in Medicine* 19, 1141–1164. <https://www.tau.ac.il/~saharon/Boot/10.1.1.133.8405.pdf>
2. Xavier Robin, Natacha Turck, Alexandre Hainard, et al. (2011) “pROC: an open-source package for R and S+ to analyze and compare ROC curves”. *BMC Bioinformatics*, 7, 77. DOI: 10.1186/1471-2105-12-77.
3. Elisabeth R. DeLong, David M. DeLong and Daniel L. Clarke-Pearson (1988) “Comparing the areas under two or more correlated receiver operating characteristic curves: a nonparametric approach”. *Biometrics* 44, 837–845.

3. Using the 607 features proposed here may not avoid identifying spurious associations. As the authors pointed out, there are many potential confounders in the presented analyses, such as staining difference, tumor purity, and histologic subtypes. However, none of these confounders were controlled or adjusted in any of the presented models, according to the reported methods. Thus, if potential spurious associations are of concern, they will likely persist in these results.

Simple models (such as generalized linear models) do not preclude or mitigate the effects of confounders. In addition, the effects of these confounders were not assessed in the study.

We thank the reviewer for this comment. We agree with reviewer #1 and the editors that this point has already been sufficiently addressed. As noted in our previous revision:

“Third, batch effects in TCGA can originate from differing tissue collection, sectioning, and processing procedures. Our validation procedure of partitioning by tissue source site does not account for all possible data artifacts, but it does control for confounding by sample collection, extraction, and other site-specific variables. Our HIF-based approach also limits the impact of spurious associations introduced by batch effects by pre-defining features based on biological phenomena.”

4. The deep learning model that serves as a baseline of comparison was not designed specifically for the task and is not tuned. As modern machine learning architectures require substantial fine-tuning, it is possible that the current baseline model is not representative of the machine learning models researchers will use for similar tasks.

We believe our deep learning model benchmarks are sufficiently representative, given that the original architectures were also designed for the task of molecular prediction. We agree with reviewer #1 and the editors that this point has been sufficiently addressed in the previous revision.

5. As pointed out by previous reviewers, the authors attempted to validate their models, but only one analysis is validated with another publicly-available dataset. The robustness of the remaining models is unknown.

We agree that the manuscript should address the limitation of external validation for the cell and tissue models and have updated the Results as follows:

“To assess model generalizability, we redeployed our BRCA cell-type model to predict cell types on H&E, FFPE WSIs from an external BRCA dataset uploaded by Peikari et al. to The Cancer Imaging Archive (TCIA)³⁵. We then repeated the same frames analysis framework using 250 frames evenly sampled across the five cell types, which revealed robust concordance between our cell-type model and pathologist consensus in these external WSIs (Supplemental Figure 4). Correlation coefficients ranged from 0.607 in macrophages to 0.926 in lymphocytes and differed from inter-pathologist correlation by a median absolute difference of 0.076. As a benchmark, inter-pathologist correlation represents the optimal performance that can be expected from models trained and evaluated using pathologist annotations as the ground truth. **External data was not publicly available for the**

remaining cancer types. While the BRCA cell-type model generalized without additional tuning, other models may require retraining when applied to new datasets.”

6. Recent pan-cancer analyses on TCGA histopathology data involved more than 20 types of cancers available from the Genomic Data Commons platform and reported the associations between oncogenes, tumor suppressor genes, and digital pathology features. Thus the analyses of the selected cancer types are not new and are covered by these prior studies.

We have reviewed the publication list from the Pan-Cancer Atlas and other projects (<https://gdc.cancer.gov/about-data/publications#/?programs=TCGA>); none conduct similar or comparable analyses to our pipeline. We believe the citations in our Introduction and Discussion sufficiently cover relevant similar works that contextualize our own. The primary contribution of our study (distinct from the pan-cancer analyses) involves prediction of molecular phenotypes with comparable performance to end-to-end models using image features that are both biologically relevant and human-interpretable.

7. The authors stated that the scale of the number of interpretable features ($m=607$) as a novelty of the study. However, many previous studies cited in the introduction section used several thousand interpretable pre-defined features via a computational framework. It is unclear if the scale is truly a highlight.

We agree with reviewer 3 that the number of interpretable features is not important. Instead, our manuscript focused on generally a set of interpretable and complex features that represent relevant biological phenomena. We agree with reviewer 1 and the editors that this point has already been sufficiently addressed.

8. The current reporting of methods is still unfortunately opaque. It is impossible for readers to reproduce the proposed features from the histopathology images or to improve the feature set. The hyperparameters and model training procedures for the cell detection model are not available either. The lack of such information makes it difficult to satisfy reporting guidelines regarding machine learning applications.

We thank the reviewer for their comment. We have added additional specificity to the Code Availability statement regarding access conditions and contents of the public repository:

“All feature tables, as well as source code for reproducing correlational analyses and molecular predictions, are available at: <https://github.com/Path-AI/hif2gene/tree/master/data/hifs>. Access to cell- and tissue-type heatmaps as well as usage of cell-type and tissue-type classification models are available upon reasonable request to academic investigators without relevant conflicts of interest who agree not to distribute the data.”

Access to heatmaps enables interested readers to recompute relevant image features and improve the feature set, while access to feature tables and related code enables validation of key findings from this study.

Reviewer #5

I am satisfied by the responses to my specific comments and do not have any further concerns regarding the publication of the manuscript.

We thank the reviewer for their help in improving the manuscript.